

**Development and performance of a high-resolution surface wave and storm surge forecast model**
**(COASTLINES-LO): Application to a large lake**
**Laura L. Swatridge[1], Ryan P. Mulligan[1], Leon Boegman[1], and Shiliang Shan [2]**
[1] Department of Civil Engineering, Queen's University, Kingston, ON, Canada, K7L 3N6.
[2] Department of Physics and Space Science, Royal Military College of Canada, Kingston, ON, Canada,
K7K 7B4.
Correspondence to: Laura Swatridge (l.swatridge@queensu.ca)
**Key Points:**
• A real-time forecast model of wind-driven hydrodynamics in Lake Ontario is developed.
• Model performance compares well with observed data and other forecast models.
• Forecast lead time impacts the accuracy of wave height and storm surge predictions.



**Abstract**

An automated real-time forecast model of surface hydrodynamics in Lake Ontario (Coastlines-LO) was developed to predict storm surge and surface waves. The system uses a dynamically coupled Delft3D – SWAN model with a structured grid to generate 48 h predictions for the lake that are updated every 6 h. The lake surface is forced with meteorological data from the High Resolution Deterministic Prediction System (HRDPS). The forecast model has been running since May 2021, capturing a wide variety of storm conditions. Good agreement between observations and modelled results is achieved, with root mean squared errors (RMSE) for water levels and waves under 0.02 m and 0.26 m, respectively. During storm events, the magnitude and timing of storm surges are accurately predicted at 9 monitoring stations (RMSE < 0.05 m), with model accuracy either improving or remaining consistent with decreasing forecast length. Forecast significant wave heights agree with observed data (1-12% relative error for peak wave heights) at 4 wave buoys in the lake. Coastlines-LO forecasts for storm surge prediction for two consecutive storm events were compared to those from the Great Lakes Coastal Forecasting System (GLCFS) to further evaluate model performance. Both systems achieved comparable results with average RMSEs of 0.02 m. Coastlines-LO is an open-source wrapper code driven by open-data and has a relatively low computational demand, compared to GLCFS, making this approach suitable for forecasting marine conditions in other coastal regions.

**1 Introduction**

Coastal regions of large lakes can face hazardous conditions with costly consequences due to strong storm events, where powerful winds generate large waves and storm surge (Danard, 2003; FEMA, 2014; Gallagher et al., 2020). Waves during these events can cause erosion, overtopping, and run-up, with the hazards being greater when the water level is elevated from storm surge. The intensity and frequency of strong storm events is increasing in the Great Lakes region as a result of climate change, as tropical storms are predicted to reach higher latitudes more often (Bender et al., 2010; Studholme et al., 2022). In addition, the mean water levels in the Great Lakes are being impacted by climate change, with large seasonal fluctuations in lake levels and record low and high water levels consistently occurring in recent years (Gronewold and Rood, 2019). The combined impacts of these projections present a greater risk for hazardous conditions in Great Lakes coastal regions, and developing better methods to understand and model the physical processes occurring during storms is important to help mitigate the risk. (Chisholm et al., 2021; Gronewold et al., 2013).






‘Real-time forecasting’ of lakes and coastal oceans can be achieved by applying numerical models to run
predictive simulations of future hydrodynamic conditions in real time. Water level, circulation, and
temperature simulations, using forecast models of large lakes and reservoirs, aid in water quality
management (Baracchini et al., 2020; Carey et al., 2021; Lin et al., 2022). Coastal hazard forecasting is also
being applied in numerous ocean regions, including the northern Gulf of Mexico where forecast systems of
water levels and waves predict hurricane impacts on various scales (Bilskie et al., 2022; Dietrich et al.,
2018; Paramygin et al., 2017). Similarly, Rey and Mulligan (2021) use a coupled Deflt3D–SWAN model
to forecast storm conditions in coastal North Carolina, investigating the influence of various atmospheric
forecast models on the results during hurricanes. Specific to lakes, the National Oceanic and Atmospheric
Administration (NOAA) has implemented forecast models for North American coastal regions, including
the Great Lakes, with the Great Lakes Coastal Forecast System (GLCFS). The GLCFS uses a high-
resolution (30 m – 2 km) hydrodynamic model (FVCOM) to simulate physical processes including currents,
temperatures, and water levels (Kelley et al., 2018; Peng et al., 2019). Waves in the Great Lakes are
predicted by Environment and Climate Change Canada's (ECCC) Regional Ensemble Wave Prediction
System (REWPS), which uses a probabilistic approach to forecast wave characteristic 3 days into the
future.

Developing deterministic forecast models that run in real-time requires dealing with the challenge of
minimizing the computational runtime of the model while still achieving accurate results (model resolution
and performance), as the forecasts must be available in advance of the actual event. In addition, clear and
efficient dissemination of forecasts must be provided to users and stakeholders. Typical real-time coastal
models require large computing resources to run high resolution and accurate forecast simulations (Bilskie
et al., 2022; Kelley et al., 2018), while fewer model applications focus on developing flexible systems that
can achieve accurate results while running on local computers, often for smaller domains, using open data
and with a smaller computational allowance (Lin et al., 2022; Rey and Mulligan, 2021).

The accuracy of numerical models for simulating the hydrodynamic response of coastal regions to storm
events has increased with advances in computing power, data availability, and the development of models
that can better represent more physical processes and their interactions, however model performance is still
limited by the quality of input and forcing data available for a simulation. Model ability also depends on
the grid resolution, with higher resolution models being more capable of resolving bathymetric features
(Bilskie et al., 2022), and the inclusion of relevant processes, such as wave-current interactions and
baroclinic effects (Asher et al., 2019; Swatridge et al., 2022). A main consideration is the accuracy of the



atmospheric forcing, as winds are the primary driver of surface behaviour, and errors in the winds translate
through as errors in the modelled results (Dietrich et al., 2018; Farhadzadeh and Gangai, 2017; Rey and
Mulligan, 2021).

A probabilistic approach can be used to account for uncertainty in atmospheric forcing by running multiple
variations of the same event, however this requires large computational resources (Baracchini et al., 2020;
Fleming et al., 2008). In deterministic forecasts of water levels in Lake Erie, error in the atmospheric forcing
was significantly larger for 240 h forecasts compared to the 120 h forecasts, which translated to increased
error in predicted water levels (Lin et al., 2022). The longer forecast predicted excessive seiching and an
underestimation in peak water level, which improved as forecast length decreased. Forecasts of hurricane
storm surge and waves in the Gulf of Mexico by Forbes et al. (2010), Dietrich et al. (2018), and Bilskie et
al. (2022) found trends of decreasing error in storm surge prediction with shorter forecast length. Longer
forecasts (~5 days) resulted in storm surge variations of up to 4 m from the best track predictions, attributed
to variability in atmospheric forcing, and for forecasts shorter than 2.5 days, simulations converged on a
solution, and error was almost constant (Dietrich et al., 2018).

The hydrodynamics of Lake Ontario have been simulated on various scales in previous studies (e.g., Huang
et al., 2010; Paturi et al., 2012; Prakesh et al., 2007; Shore, 2009). Numerical models have also been used
to simulate waves and circulation during extreme events in the Kingston Basin (Cooper and Mulligan, 2016;
McCombs et al., 2014a; McCombs et al., 2014b). Sogut et al. (2019) used a combination of analyzing
historical water level and wave data, as well as numerical modelling of extreme storm events to gain insight
on lake seiching, storm surges, and wave patterns. Historical data have also been studied to determine the
risk of flooding due to storm surge along the Lake Ontario shoreline with a statistical model (Steinschneider,
2021). Surface waves and storm surge were simulated over the entire lake by Swatridge et al. (2022) during
recent storm events. Their study investigated the influence of different wind fields on the accuracy of storm
surge simulation, finding that variations in meteorological forcing were the primary source of uncertainty
in model results.

In the present study, an existing depth-averaged numerical model of Lake Ontario (Swatridge et al., 2022)
was applied to the lake to forecast water levels and waves in real-time, driven by spatially varied wind
fields from a high-resolution wind forecast model. The workflow develops an open-source Python- and
MATLAB-based wrapper code, that has been successfully applied to other systems using different
hydrodynamic models as part of the Canadian Coastal and Lake Forecasting Model System (Coastlines;
https://coastlines.engineering.queensu.ca; Lin et al., 2022; Rey and Mulligan, 2021). This flexible



methodology uses open access forcing/validation data and requires a relatively low computational demand,
compared to other existing Great Lakes storm surge models, allowing for application to other locations.
Model performance is evaluated by comparing results to near-real time observed data. Forecast results, for
storm surges and waves are statistically investigated over forecast lead times ranging from 6 to 48 h.

**2 Methods**

*2.1. Modelling Approach*
A two-dimensional (depth-averaged) coupled hydrodynamic-wave model is applied to Lake Ontario to
simulate wind driven hydrodynamics and waves using Delft3D-SWAN. The Delft3D flow model calculates
non-steady flow on a structured grid by solving the Reynolds-Averaged Navier Stokes equations (Lesser et
al., 2004). Wave conditions are simulated with the phase-averaged wave model, Simulating WAves
Nearshore (SWAN), which uses the spectral action balance equation to compute random wind-generated
waves. SWAN accounts for non-linear wave interactions, wave propagation, refraction, dissipation due to
whitecapping, bottom friction and depth-induced breaking (Booij et al., 1999). The models are dynamically
coupled to account for wave-current interactions. Radiation stress gradients from SWAN simulations are
input into the horizontal momentum equations in Delft3D to account for the impacts of waves on
circulation, such as wave-induced mass fluxes driving currents, and enhanced bed shear stress. Results from
the hydrodynamic simulation are then used to update water levels and circulation in the wave model.

Model setup choices were made based on simulations by Swatridge et al. (2022) which were adapted for
the present study to minimize computational demand, allowing the system to run in real-time. The Delft3D
simulation uses a curvilinear grid with a horizontal resolution gradually ranging from 250-450 m, with
higher resolution in nearshore areas, and a coarser grid with resolution ranging from 350-600 m for the
wave model. Flow simulations are depth-averaged and barotropic, shown by Swatridge et al. (2022) to
accurately represent surface storm surge in Lake Ontario, with root mean squared errors (RMSEs) between
observations and model results ranging between 0.01 m - 0.07m during several major events. Bathymetry
data was interpolated to the grid from the US National Centers for Environmental Information's (NCEI) 3-
arcsecond (~ 90 m) resolution dataset with supplementary data from the ETOPO1 global relief model with
a resolution of approximately 1.3 km (Fig. 1). Detailed sensitivity testing for this model was completed in
Swatridge et al. (2022) to calibrate model parameters. Simulations use a time step of 120 s to satisfy the
Courant–Friedrichs–Lewy stability criterion and coupling between the flow and wave models occurs every
60 minutes.


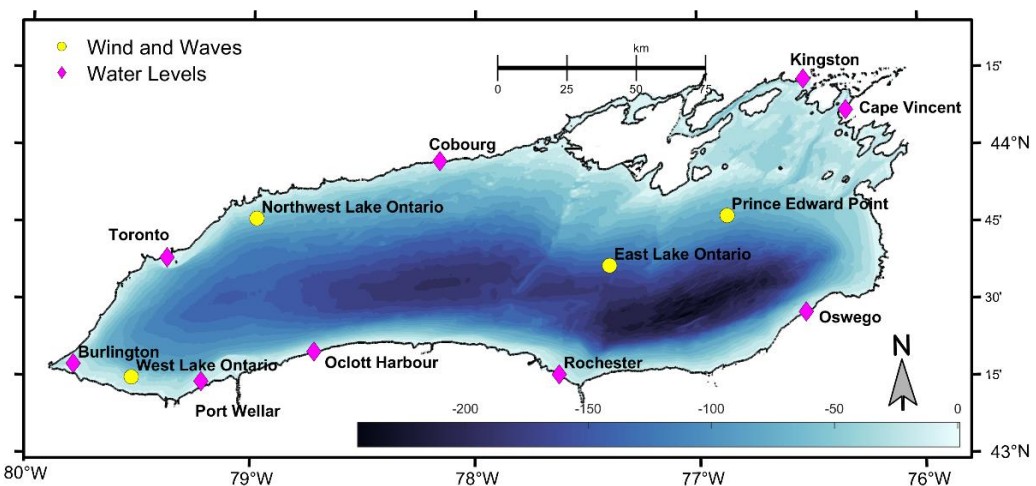


**Figure 1**: Map of Lake Ontario showing bathymetry and the location of real-time water level, wind, and
wave observation stations.

Spatially varied atmospheric input from the Meteorological Service of Canada (MSC) High Resolution
Deterministic Prediction System (HRDPS) is used to drive the model (Milbrandt et al., 2016). HRDPS is
an hourly assimilated forecast system downscaled from the larger scale Regional Deterministic Prediction
System (RDPS) that provides hourly predictions of surface pressure and wind velocity components with a
horizontal resolution of 2.5 km for the pan-Canada domain. The system runs every 6 h, predicting
atmospheric conditions 48 h into the future. This wind-forcing was successfully used by Swatridge et al.
(2022) to simulate the lake surface response to a range of storm conditions. Their modelled results for water
levels and surface waves agreed with observations at up to 16 locations in Lake Ontario, resulting in
maximum difference between predicted and observed peak wave heights and water levels of 0.4 m and
0.08 m, respectively. No lateral open boundary conditions are applied to account for inflows and outflows
to the lake, as previous work has found the major riverine flows (Niagara and St. Lawrence Rivers) have a
negligible hydrodynamic influence on large-scale circulation and water levels over event-based timescales
(Prakash et al., 2007).

*2.2. Development of an Automated Prediction System*
The forecast system uses a combination of code written in MATLAB and Python to automatically run every
6 h and has been operational since May 2021 (https://coastlines.engineering.queensu.ca/lake-ontario/). The
workflow (Fig 2) consists of pre-processing, model simulation and post processing stages. For pre-





processing, the system is initiated when a new HRDPS forecast becomes available. Python is used to
download the latest forecast and MATLAB is used to automatically process the atmospheric forcing and
write input files for Delft3D-SWAN. The Delft3D model definition files are then updated with the correct
time information.

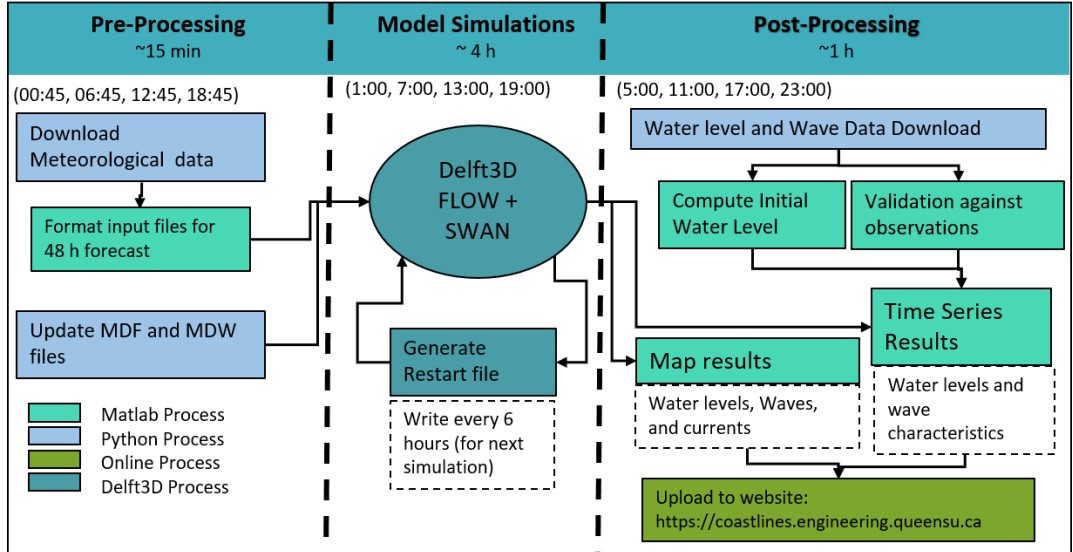


**Figure 2.** Diagram of the automated workflow for processes performed for each model cycle (every 6 h

initiated by Windows Task Scheduler) on the local Coastlines server.


Model simulations cover a period of 48 h and are 'hot-started' with a restart file from a previous model run
if available. If a restart file is not available, simulations begin from rest with initial water levels of 0 m and
current speeds (u) of 0 m s$^{-1}$ throughout the lake. When the simulation finishes, all available real-time
observed data, outlined in Table S1 in the supplementary material, is downloaded using Python, which is
then processed in MATLAB. Observed water levels, at each station, are averaged over the previous 12 h
and used to locally adjust the datum of the model outputs. We acknowledge that assimilating observed
water levels into the initial conditions may be a preferred approach, but this is beyond the scope of the
present study and may be incorporated into future versions on Coastlines-LO. The model simulates high
frequency variability in water levels generated by winds. Seasonal changes in water levels due to inflows,
outflows, and evaporation are not included, but are accounted for in post-processing.

Time series plots of observed water levels and wave heights are automatically compared to the forecast
model results from the previous 2.5 days at the observation locations and additional plots are created to
provide predictions at other locations of interest with no observed data (Fig. 1). Spatial snapshots of model



results across the lake are generated at select times, as well as animations showing key output parameters
during the forecast simulation. All outputs are exported to Google Sheets and displayed on the project
webpage, https://coastlines.engineering.queensu.ca/. The system runs in a Windows environment using 16
cores of a 32-core XEON workstation, with each workflow cycle taking approximately 5 h to complete a
48 h forecast simulation.

*2.3. Real-time Comparison between Model Results and Observations*
Near real-time observations of water surface elevation ($\eta$) data are available at 9 water level gauges in Lake
Ontario from the National Oceanic and Atmospheric Administration (NOAA) and the Department of
Fisheries and Oceans Canada (DFO), with temporal resolutions of 3 minutes and 6 minutes, respectively
(Fig. 1; Table 1). Hourly surface waves and winds are measured in Lake Ontario at one US National Data
Buoy Center (NDBC) buoy and ECCC buoys from spring to early winter (these buoys are removed in
winter due to the possibility of ice damage). The buoys report the significant wave height ($H_s$), peak wave
period ($T_p$), surface wind speed and direction averaged over an 8-minute period (U).



**Table 1:** List of real-time observed data sources in Lake Ontario

| Name | Longitude | Latitude | Depth | Parameter | Source |
|---|---|---|---|---|---|
| Prince Edward Point | -76.87 | 43.78 | 68 m | Wave; Wind | ECCC |
| West Lake Ontario | -79.53 | 43.25 | 35 m | Wave; Wind | NDBC |
| Northwest Lake Ontario | -78.98 | 43.77 | 54 m | Wave; Wind | NDBC |
| East Lake Ontario | -77.40 | 43.62 | 140 m | Wave; Wind | NDBC |
| Oswego | -76.52 | 43.46 | N/A | Water Level | NOAA |
| Rochester | -77.63 | 43.27 | N/A | Water Level | NOAA |
| Olcott Harbour | -78.72 | 43.34 | N/A | Water Level | NOAA |
| Cape Vincent | -76.33 | 44.12 | N/A | Water Level | NOAA |
| Port Wellar | -79.22 | 43.24 | N/A | Water Level | DFO |
| Cobourg | -78.16 | 43.96 | N/A | Water Level | DFO |
| Burlington | -79.79 | 43.29 | N/A | Water Level | DFO |
| Kingston | -76.52 | 44.22 | N/A | Water Level | DFO |
| Toronto | -79.38 | 43.64 | N/A | Water Level | DFO |


For long term analysis of results, the residual component of the water level data, representing storm surge,
is isolated at the gauge locations by finding the difference between the total water level and the average
water level, calculated using a gaussian window of 7 days (Steinschneider et al., 2021). Model performance
is quantified by computing error statistics, including the RMSE, normalized RMSE (NRMSE), and the
correlation coefficient (r). Strong storm surge events are identified from the water level data using the
peaks-over-threshold method (Steinschneider et al. 2021). Forecast error, during select events, was
evaluated by computing error metrics for consecutive forecasts leading up to the peak of the event. For each
forecast, the relative error (RE), between observed and simulated maximum storm surge or wave heights,
was computed, and the RMSE was computed over a 6 h period that included the peak of the event.

**3 Results**

*3.1. Long-term model performance*
Simulation results, for water levels and waves, at the observation locations, were compiled over the 20-
month operational period. The first 6 h of each 48 h forecast were stitched into a single time series, and
these results were compared to the observed data (Fig. S1 in the supplementary material). During this time,
seasonal changes in the observed mean lake level fluctuated by over 1 m, with the highest water levels



occurring in May 2022. The ability of the model to reproduce storm surge was investigated over a four-
month period when multiple storm events occurred (106 days from 15 September 2022 to 30 December
2022; Fig. 3). Stations with larger ranges of observed water levels (i.e., Burlington, Cape Vincent), located
at the east and west ends of the lake (i.e., Fig. 3c, g, i) show a slight bias, where the model tended to slightly
overpredict maximum and minimum values, corresponding to larger RMSE values (Table 2). These stations
also tended to show a stronger correlation (r = 0.83 – 0.86); whereas observation points with typically
smaller ranges in water levels (Fig. 3a, c) resulted in weaker correlations (r = 0.76 – 0.79). Normalized
results show comparable error statistics at all stations, with larger errors occurring at locations with smaller
storm surges (i.e., Rochester, Oswego).

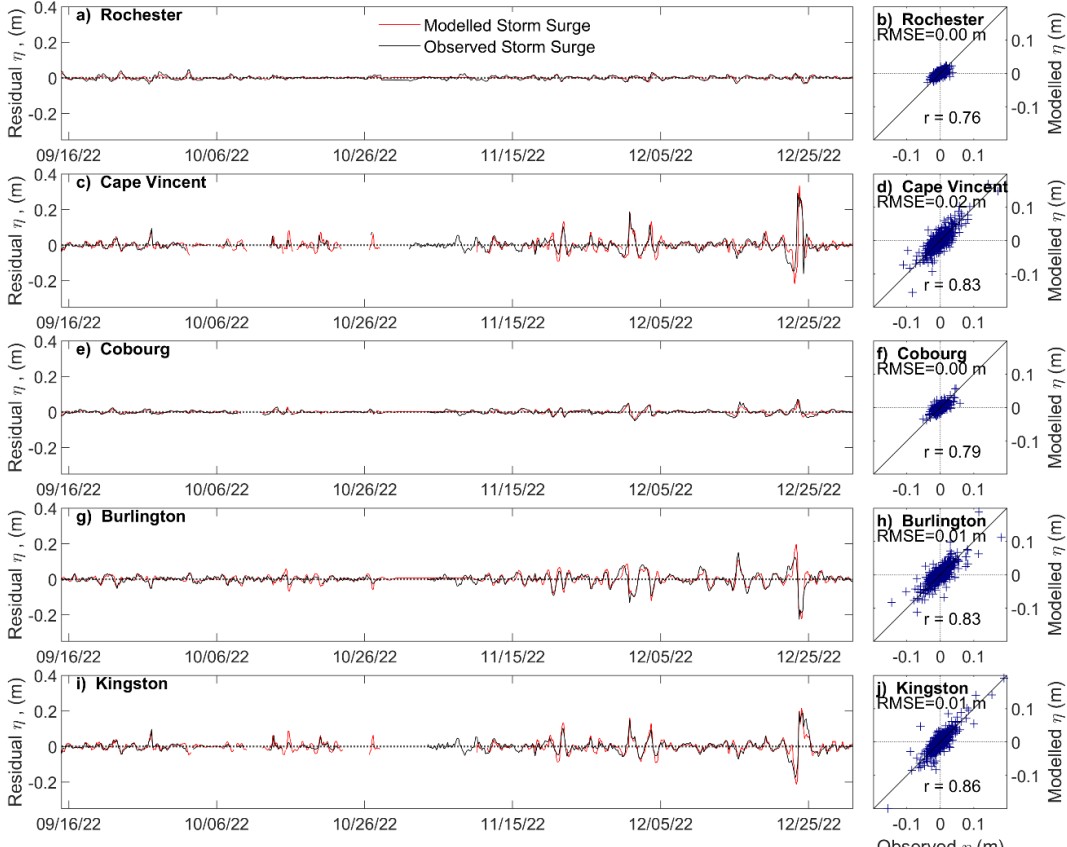


**Figure 3:** Observed (black) and modelled (red) residual water levels at select observation points over a 3
month period (September – December 2022) with corresponding scatter plots and error statistics over this
period at select locations.



**Table 2:** Error Statistics for residual water level results over 106 days (September 15 – December 30, 2022)

|  | Minimum η (m) | Mean η (m) | Maximum η (m) | RMSE (m) | NRMSE (m) | r |
|---|---|---|---|---|---|---|
| Oswego | -0.10 | 0.07 | 0.12 | 0.01 | 0.15 | 0.80 |
| Rochester | -0.03 | 0.03 | 0.04 | 0.00 | 0.16 | 0.76 |
| Olcott | -0.16 | 0.04 | 0.11 | 0.01 | 0.19 | 0.80 |
| Cape Vincent | -0.22 | 0.10 | 0.34 | 0.02 | 0.16 | 0.83 |
| Port Wellar | -0.19 | 0.06 | 0.16 | 0.01 | 0.14 | 0.86 |
| Cobourg | -0.08 | 0.04 | 0.07 | 0.01 | 0.14 | 0.79 |
| Toronto | -0.16 | 0.07 | 0.14 | 0.01 | 0.14 | 0.83 |
| Burlington | -0.22 | 0.10 | 0.20 | 0.02 | 0.14 | 0.83 |
| Kingston | -0.21 | 0.09 | 0.25 | 0.01 | 0.14 | 0.86 |


Results for simulated $H_s$ over the 600-day operational period at buoy locations show the largest waves
occurred during winter, between December and March (Fig.4). During this time, no monitoring data was
available for comparison and Lake Ontario could potentially experience partial ice-cover in nearshore areas,
impacting the wave environment (Anderson et al., 2018). Stations in the northeast region of the lake (Prince
Edward Point, East Lake Ontario) generally experienced the largest waves, due to the prominent north-
easterly direction of storms over the lake resulting in a larger fetch at these locations. Error statistics show
similar values for RMSE at these points however Prince Edward Point had the lowest correlation coefficient
(Fig. 4a, b; r = 0.71), while East Lake Ontario showed the highest correlation (Fig. 4c, d; r = 0.88). Lower
RMSE were at stations with smaller waves (Fig. 4e, g), and normalized results (Table 3) show comparable
results at all buoys (NRMSE = 0.42 – 0.53 m).



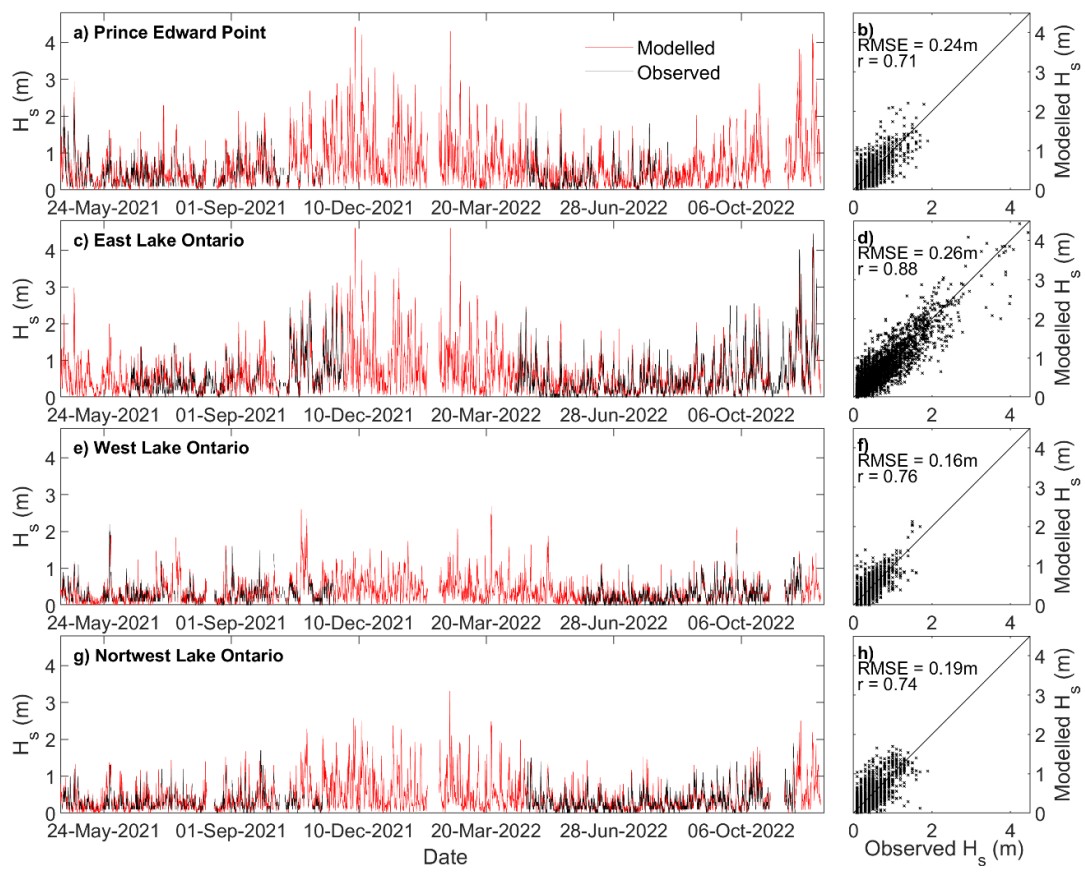

**Figure 4:** Time series of observed (black) and modelled (red) significant wave height over the duration that the buoys were in the lake (September -December 2022) with corresponding error scatter plots at the location of the 4 buoys.



**Table 3:** Error statistics for significant wave heights at the buoy locations over 600 days (April 21, 2021 –
December 12, 2022)

| Location | Mean $H_s$ (m) | Maximum $H_s$ (m) | RMSE (m) | r | NRMSE (m) |
|---|---|---|---|---|---|
| Prince Edward Point | 0.44 | 3.82 | 0.24 | 0.71 | 0.53 |
| East Lake Ontario | 0.62 | 4.42 | 0.26 | 0.88 | 0.42 |
| West Lake Ontario | 0.34 | 2.60 | 0.16 | 0.76 | 0.48 |
| Northwest Lake Ontario | 0.35 | 2.29 | 0.19 | 0.74 | 0.53 |

*3.2. Storm event forecasts*

The performance of the model was evaluated during an event on November 11, 2021. During this event, wind speeds reached up to 15 m s$^{-1}$, with the direction rotating clockwise from the southeast to the west over a 24 h period. Overlapping 48 h HRDPS forecasts (i.e., generated every 6 h) were validated against buoy observations, with good agreement found between modelled and predicted total wind speeds and directions, with peak wind speeds underrepresented by at most, 4.21 m s$^{-1}$ at Northwest Lake Ontario and overpredicted by up to 2.61 m s$^{-1}$ at Prince Edward Point (Fig. S2 in the supplementary material)

This event resulted in an observed storm surge of up to 0.16 m in the northeast region of the lake, at Cape Vincent and Kingston. The forecast simulations captured the timing and magnitude of the event peak, with predicted surge values ranging between 0.12 m – 0.17 m (Fig.5d, i). A set down of about 0.10 m was recorded at the Burlington station, which was underpredicted by the model by up to 0.05 m. The simulated results at this location predicted water levels up to 0.05 m higher than the observations for the 24 h preceding the storm (Fig.5h). Notable error can also be identified at Cobourg (Fig. 5f) with the model predicting negligible fluctuations in the water surface, but observations show some oscillations (0.05 m).

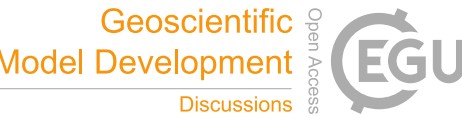

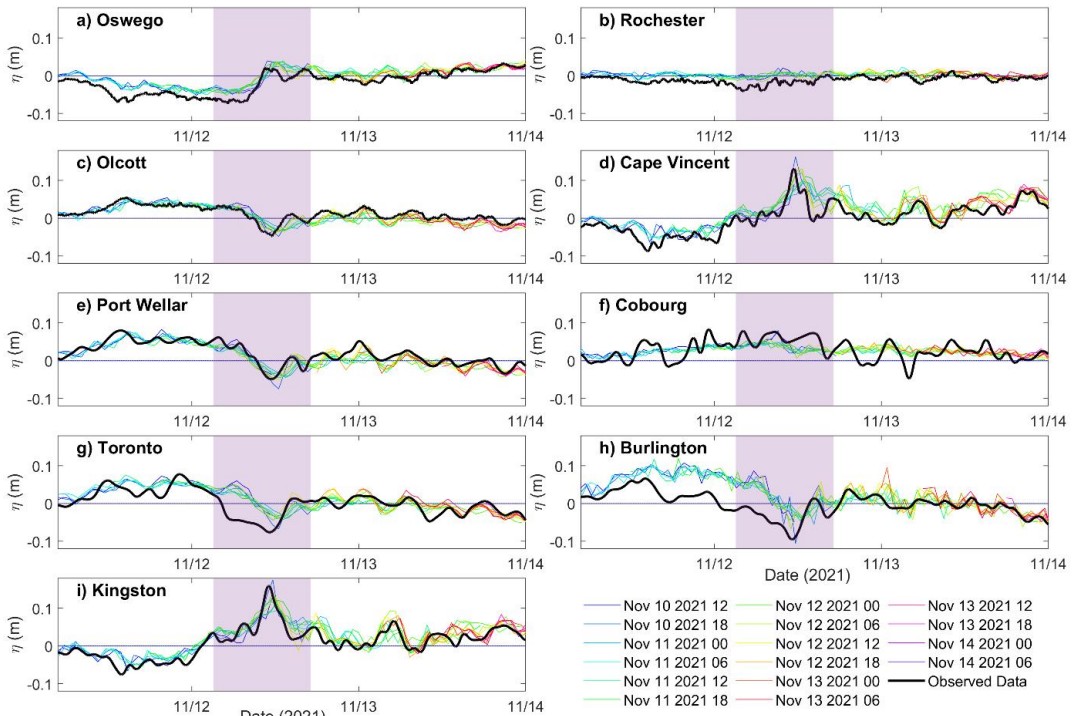

**Figure 5:** Time series of measured water levels at various observation points compared to forecasted data
from progressive model simulations. The highlighted area indicates the time over which error statistics are
computed.

Forecast performance was quantified by computing error statistics, over the duration of the event, for each
forecast leading up to the time of peak water level. The largest errors occurred at the location of the set
down, Burlington and Toronto, with a nearly constant RMSE of 0.03 m, and RE of 12% and 10%
respectively (Fig. 6c, d). The errors at all stations remained fairly constant with RMSE and RE under 0.03 m
and 10%, respectively, for each new forecast. However, map results showing the spatial variability in water
level predictions from forecasts 12 h and 36 h before the storm peak show large differences (Fig. 6a,b). The
earlier results (Fig. 6a) simulated a far less extensive storm surge in the northeast region of the lake than
what was subsequently predicted 24 h later (Fig. 6b), when the storm surge was simulated to impact most
of the northeast shoreline. The later forecast also predicted spatially larger set-down, about 0.10 m more
than the earlier forecast in the western region of the lake.



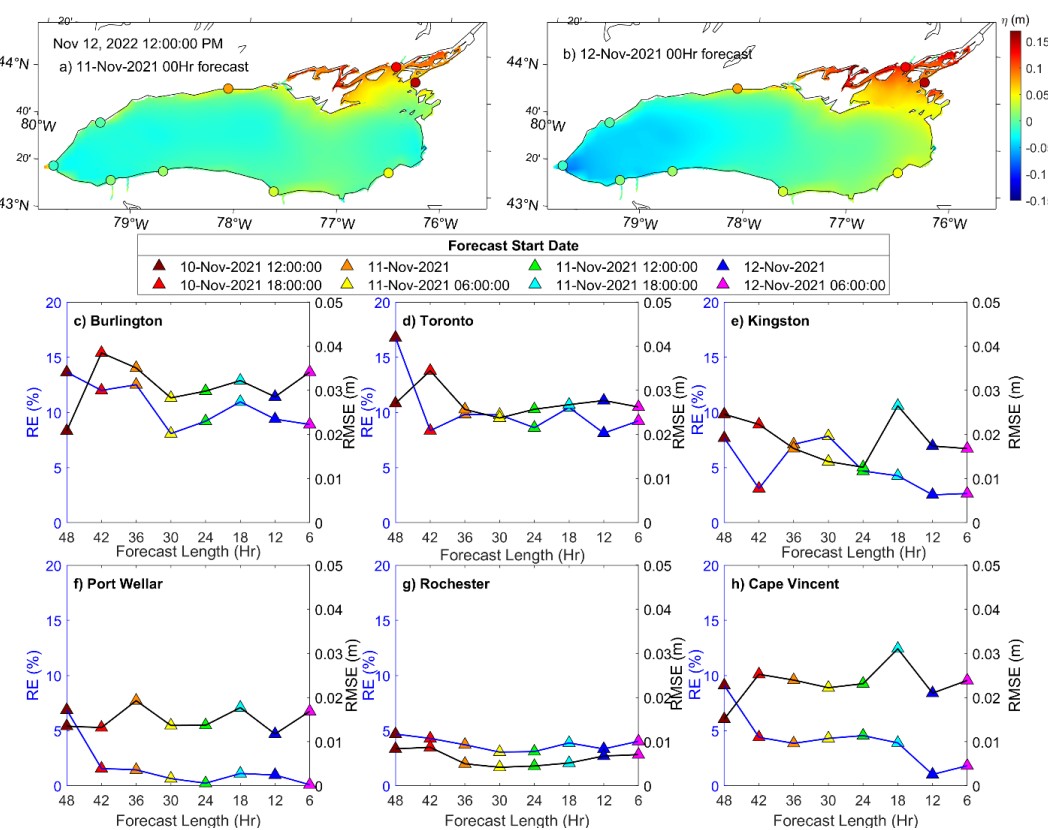

**Figure 6:** Contour plots showing maps of modelled water levels at the peak of the storm event from two different forecasts, starting a) November 11, 00:00 UTC and b) November 12, 00:00 UTC with observed data plotted at the observation locations in black circles. Panels c) to h) show metrics including the Relative Error (RE) and Root Mean Square Error (RMSE) for peak storm surge magnitude at the locations of 6 selected water level gauges from the 8 forecasts preceding the storm event.

Measured waves during this event reached up to 2.10 m, with the buoys in the western region of the lake (Fig. 7c, d) experiencing peak wave heights about 12 h earlier than the buoys in the eastern region of the lake (Fig. 7a, b), due to the shift in wind direction during the storm. Overall, forecast simulations captured the magnitude of the waves all stations, with some error, and approximately 5 h delay in the timing of the peak $H_s$ at Prince Edward Point (Fig. 7a). Error for waves during this event, at all stations, was constant for consecutive forecasts at all stations, with RMSE between 0.03 – 0.25 m and RE between 1-12%. Despite the generally consistent results, at the buoy locations, maps from different forecasts show distinct changes between the 36 h forecast (Fig. 8a) and the 6 h forecast (Fig. 8b). Simulated wave fields in the northeast

region of the lake showed similar results between forecasts, but in the northwest, predicted wave
magnitudes and directions were distinctly different. The earlier forecast predicted waves under 0.70 m
coming from the southeast, whereas the later forecast showed larger waves ($H_s = 0.50 - 1.00$ m) from the
southwest, which can be attributed to changes in forecasted wind-fields.

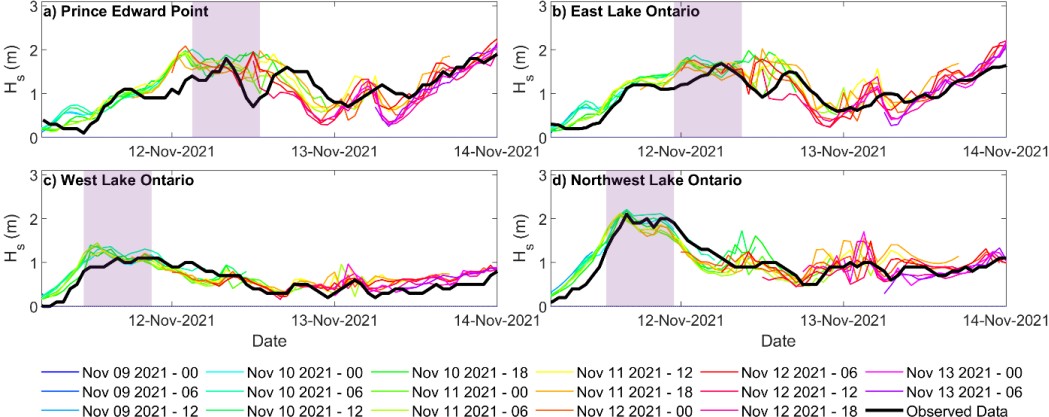

**Figure 7:** Time series of measured $H_s$ at the location of the 4 buoys compared to modelled data from
progressive model forecasts for Event 1 (November 12, 2021). The highlighted area indicates the time over
which error statistics are computed.

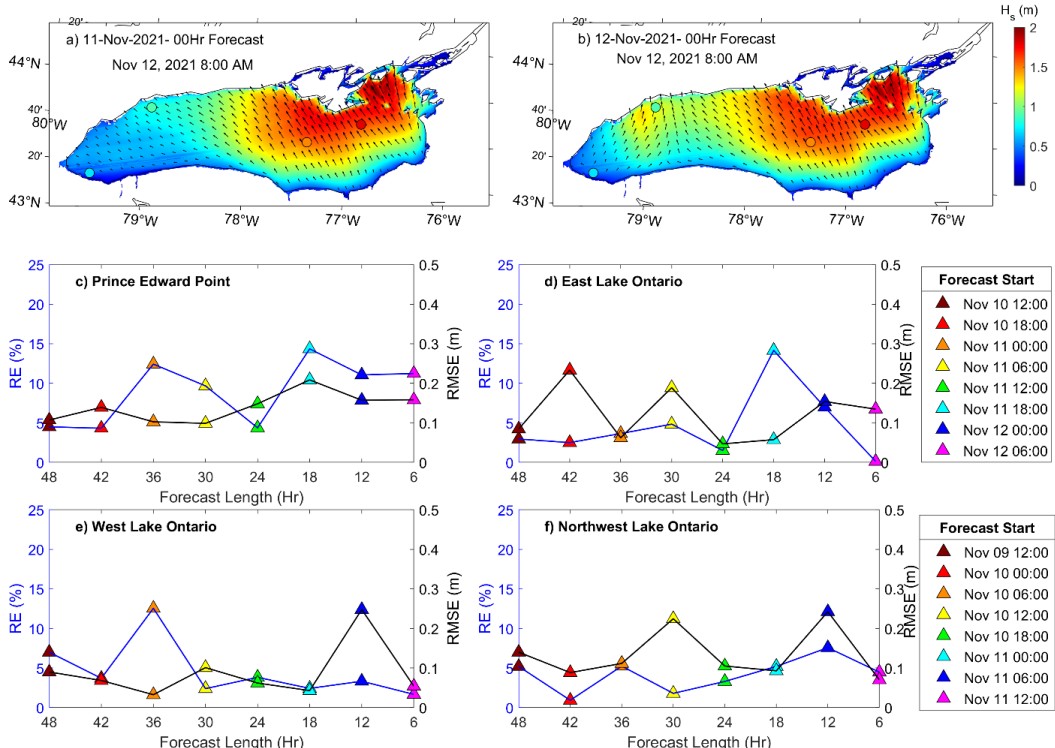

**Figure 8:** Contour plots showing maps of modelled waves at the peak of the storm event from two forecasts, starting a) November 11, 00:00 UTC and b) November 12, 00:00 UTC with observed data plotted at the observation locations in black circles. Panels c) to f) show metrics including the Relative Error (RE) and Root Mean Square Error (RMSE) for significant wave height at the locations of 4 buoys from the 8 forecasts preceding the storm event on November 12, 2021, 12:00 UTC.

For further investigation into model performance during storm events, wave forecasts during the event that resulted in the largest observed wave heights (December 1, 2022, Fig. 3c) were examined. During this storm, the lake experienced sustained easterly winds for almost 24 h, reaching speeds > 20 m s$^{-1}$ on December 1, 14:00 UTC, generating waves > 4 m (Fig. 9. Data was only available from the one buoy at East Lake Ontario during this event, which recorded a maximum H$_s$ = 4.46 m. The forecasts initially underestimated this value, with a maximum predicted wave height of H$_s$ = 4.19 m from the forecast starting on November 29 18:00 UTC, and the next forecast then overestimated this value (H$_s$ = 4.54 m). Subsequent forecasts slightly underestimated the peak value, with the lowest predicted peak H$_s$ = 4.26 m and the maximum values occurring ~1 h after the observed peak. All forecast results tended to overestimate the



peak wave period, with predicted values ranging between 7.8 - 8.1 s, compared to an observed maximum
value of 7.2 s.

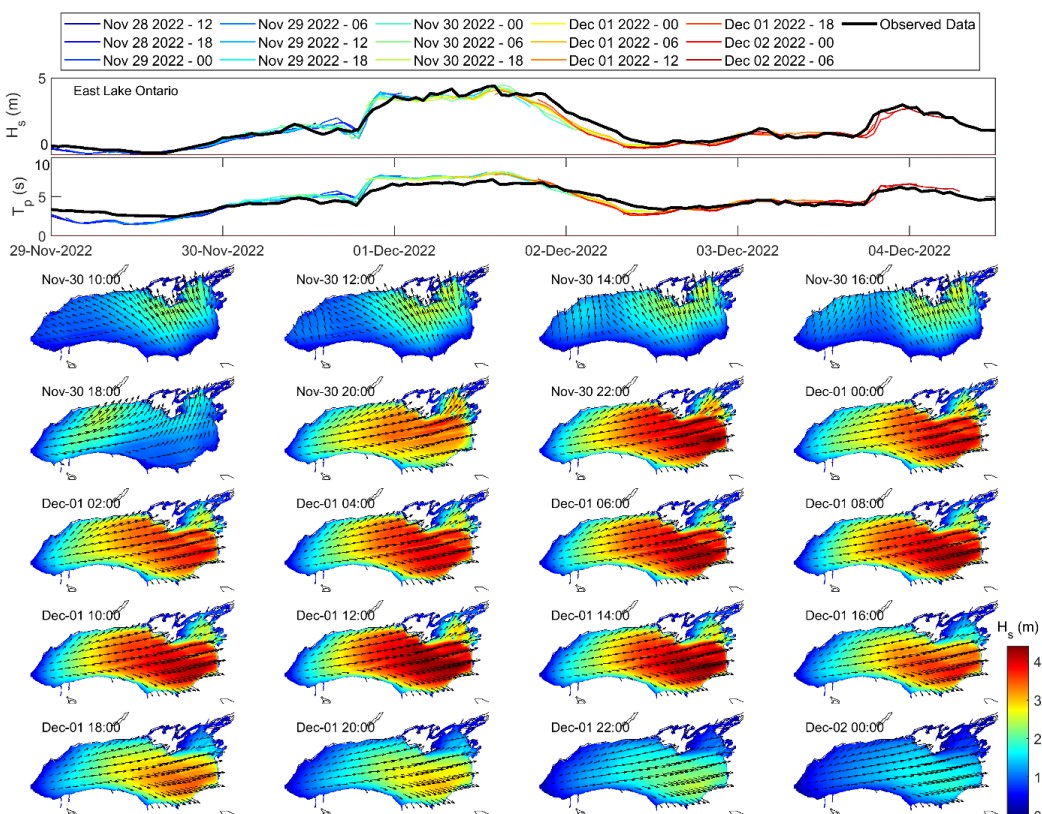


**Figure 9:** Variability in significant wave height during a storm event: measured $H_s$ compared to progressive
forecast results at the Prince Edward Point Buoy for Event 3 (December 1, 2022; top) and maps of $H_s$ and
wave direction shown at an interval of 2 h (every 10th vector is shown for clarity).

**4 Discussion**

*4.1. Forecast Lead Times*
Water level forecasts during a storm event on December 8, 2021, were examined in relation to forecast lead
time. During this event, 21 m s$^{-1}$ winds generated a storm surge of approximately 0.20 m along the northeast
coast, and a resulting setdown of 0.10 m on the opposite end of the lake. Error statistics throughout the peak





of the event, as a function of forecast lead time, were plotted at select stations (Fig. 10). The timing and
magnitude of the storm surge was well represented by the forecast model, with RMSE < 0.05 m for all
forecasts and a maximum RE =14%.

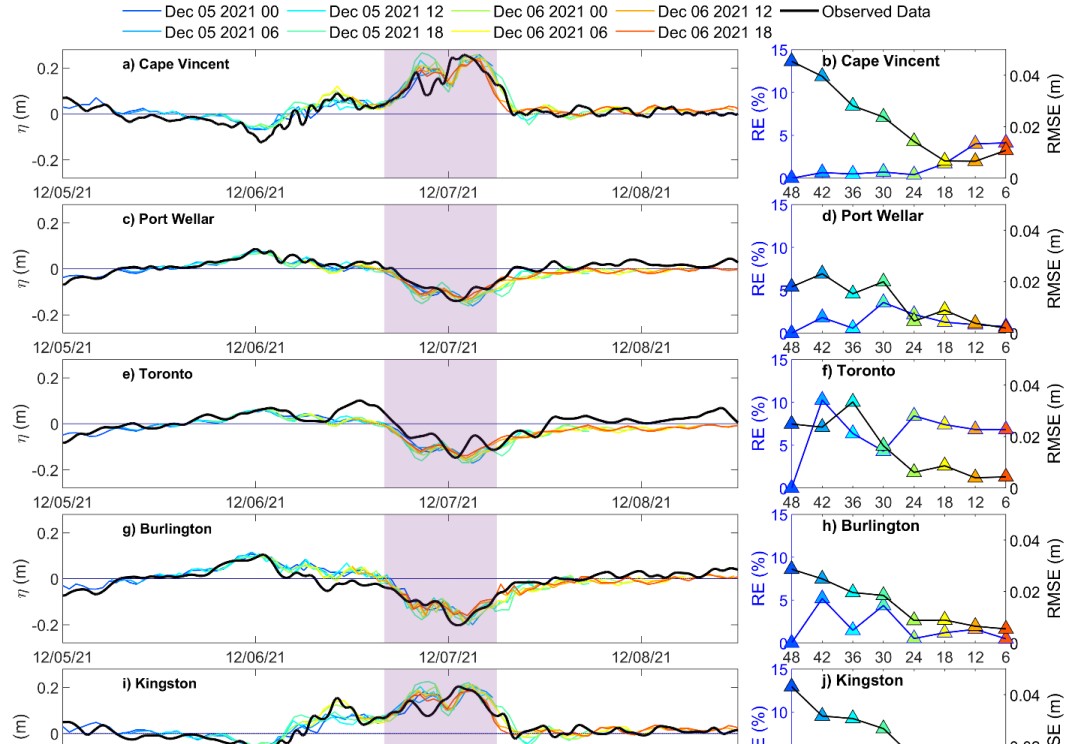


**Figure 10:** Time series of measured water levels at select observation points compared to forecasted data
from progressive model simulations for Event 3: December 08, 2021, with corresponding plots showing
computed RMSE calculated over the shaded area and percent error in peak storm surge from the 8 forecasts
preceding the storm event.

Trends in the error can be identified for this event at all stations, with notable patterns corresponding to
locations with larger fluctuations in water level (i.e., Cape Vincent, Kingston, Burlington). At these sites,
forecast error tended to decrease as the forecast length shortened. At Cape Vincent, the initial 48 h forecast
had an RMSE of 0.05 m and by the 18 h forecast, the RMSE had decreased to 0.01 m. However, after the
18 h forecast there was a slight increase in RE from less than 1% to about 5% (Fig. 10b). Trends in





decreasing error were also observed at Kingston, where a similar decrease in RMSE was observed, and the
RE was maintained between 1 – 5%, corresponding to a maximum underprediction of about 0.05 m (Fig.
10i, j). The locations with smaller ranges in surface fluctuations (Toronto, Port Wellar) generally showed
constant error (0.02 m and ~1% at Port Wellar; 0.01 m and 7% at Toronto) for consecutive forecast results
over the duration of this event (Fig. 10d, f).

Hydrodynamics in the model are only driven by atmospheric forcing, which is a primary source of
uncertainty in simulations of surface dynamics in large lakes. The accuracy of meteorological forecasts
typically decreases with increasing length due to assimilation schemes using observations and satellite
imagery to yield more accurate results (Buehner et al., 2015). Therefore, it is expected that hydrodynamic
forecast simulations will increase in accuracy as the lead time to a storm event decreases. For forecasts of
storm surges in other Great Lakes (e.g., Lake Erie; Lin et al., 2022) and coastal seas (e.g., Gulf of Mexico;
Dietrich et al., 2018), improvements in storm surge predictions are directly linked to increased accuracy in
meteorological forcing leading up to an event. However, our Lake Ontario model results do not follow a
consistent trend between different events, either improving (Fig. 10) or maintaining accuracy (Fig. 6;
Fig. 8). Despite model accuracy being constant at the observation locations, changes in the spatial
variability of predicted water levels and wave conditions for different forecasts are not clearly
communicated through time series analysis but are qualitatively shown in maps of results (Fig. 6; Fig. 10).


*4.2. Comparison with Other Models*
The current work (Coastlines-LO) makes use of a relatively simple, low computational demand modelling
approach. The performance of this model can be compared to the GLCFS, which delivers a higher resolution
and more complex forecast system in throughout the Great Lakes. Differences between these models can
be explained according to fundamental differences in the setup of each system, including different
hydrodynamic models, grid resolutions, and atmospheric forcing inputs. The GLCFS uses the 2 km
horizontal resolution High Resolution Rapid Refresh (HRRR) meteorological forcing, which is comparable
to HRDPS (2.5 km), however previous studies have found that wind and direction predictions can vary
between these models (Rey and Mulligan, 2021; Swatridge et al., 2022). The inclusion of waves in the two
systems is also accounted for differently, with a separate model (WaveWatch III) used to simulate waves
in the GLCFS, while Coastlines-LO uses a dynamically coupled wave and flow model that accounts for
wave-current interactions. The inclusion of wave coupling in simulations of the Great Lakes can impact
water level predictions (Mao and Xia, 2017). The GLCFS runs on NOAA's high performance computing
system, and the larger computational power allows it to include 3D baroclinic processes while still running




396 in the required timeframe, whereas the Coastlines-LO system in the present study uses a 2D, depth averaged

397 approach, and therefore doesn't resolve vertical gradients in lake temperature or 3D circulation. The

398 inclusion of river inflows and outflows in the GLCFS also allows the model to simulate seasonal changes

399 in the mean lake water level instead of accounting for these changes based on observed data in post-

400 processing.

402 Forecasts results from both models were compared to observed data over a 6-day period in December 2022,

403 during which 2 storm events occurred (Fig. 11). Results from the first 6 h of subsequent forecasts are

404 combined to construct a water level time series at observation points for both models for the entire duration.

405 Both models represent trends in water levels over this, resulting in comparable metrics, with an average

406 RMSE 0.02 m for both models, and r = 0.73 and 0.74 for Coastlines-LO, and GLCFS, respectively. GLCFS

407 achieved better predictions of peak water levels at Oswego for the event on December 1(RE = 30% for

408 GLCFS, RE = 51% for Coastlines-LO; Fig. 11a), and more accurately represented the surface fluctuations

409 observed over the entire 6 day period at Toronto (Fig. 11f).While GLCFS was able to represent water levels

410 at some locations, Coastlines-LO had higher accuracy predictions at others (Fig. 11c, d). At Port Wellar

411 and Cape Vincent, Coastlines-LO better predicted the peak set-down and set-up on December 1 by 0.01 m

412 and 0.03 m respectively, while GLCFS underpredicted at these locations by 0.05 m and 0.09 m. Boths

413 models had difficulty simulating the second storm surge (December 3) at Oswego and Cape Vincent (Fig.

414 11 a, c), where the observed surge occurs approximately 3 h before the predicted peak. At the Kingston

415 station (Fig. 11h), storm surges of 0.25 m and 0.30 m are observed. Coastlines-LO yielded better predictions

416 for the first event, simulating a peak value of 0.24 m, compared to 0.28 m predicted by GLCFS, while

417 GLCFS performed better for the second event, with a predicted storm surges of 0.28 m and 0.22 m for

418 GLCFS and Coastlines-LO, respectively. Therefore, while the GLCFS offers several advantages,

419 Coastlines-LO has the benefit of a low computational demand and usage of the flexible open-source

420 wrapping code and that allows for easy adaption to include different hydrodynamic models and investigate

421 different field sites (e.g., Lin et al., 2022; Rey and Mulligan 2021), while still achieving very comparable

422 results simulating short term water level fluctuations in Lake Ontario.





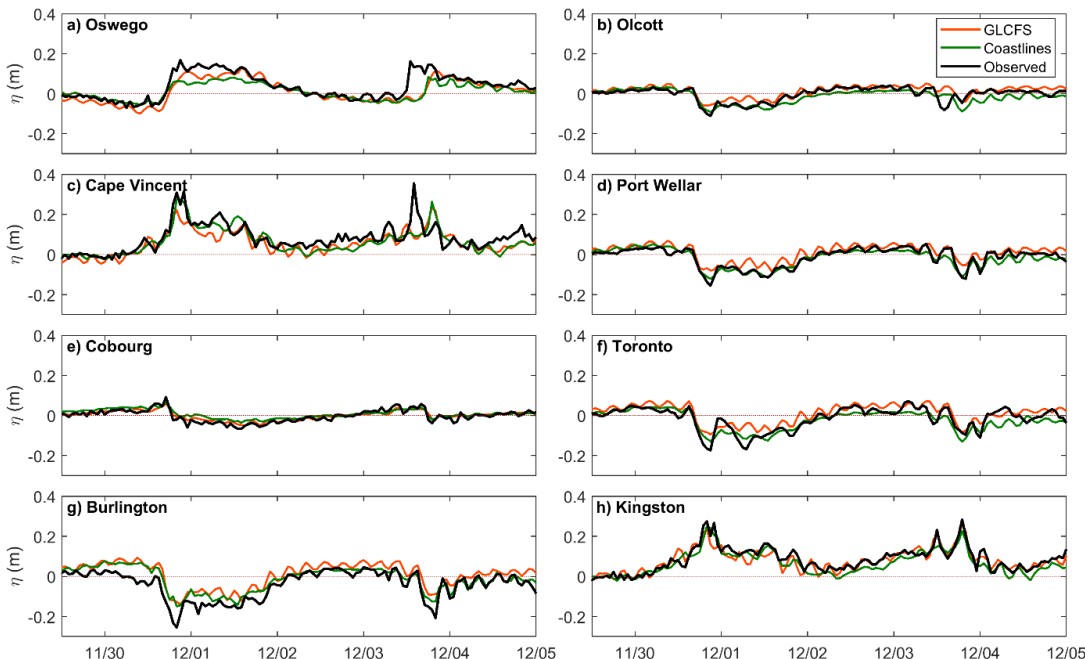

**Figure 11:** Compiled Coastlines-LO forecast results compared to forecasts from the GLCFS and observed
data at select water level gauge locations interpolated to a 30 minute time resolution for 2 subsequent events
between November 30 – December 5, 2022.

*4.3. Limitations and Uncertainties*

Sensitivity testing and calibration of the numerical model this system is based on, comes from the work of
Swatridge et al. (2022), which found that 3D simulations of Lake Ontario improved predictions of surface
behaviour compared to 2D depth averaged simulations. The 3D simulation allowed the model to account
for transfer of surface momentum into baroclinic motions, giving a better representation of current
velocities and surface seiching following a storm event, resulting in reduced RMSE during storm events by
up to 12%, and improvement in modelled peak storm surge magnitude by up to 0.03 m. While 3D
simulations improved accuracy, they also increased the computational runtime of a 24 h simulation from
about 2.5 h to 4 h. Ten-day forecasts of 3D hydrodynamic processes in Lake Erie has been achieved by Lin
et al. (2022) in using the AEM3D model with a similar Coastlines computational workflow as the current
work; however, the Lake Erie model in on a coarser 2 km horizontal grid and does not couple with SWAN
to predict surface waves, which is computationally expensive compared to hydrodynamic simulations.
Therefore, to apply this model in real-time with a new simulation every 6 h, 2D simulations are used,
potentially resulting in up to 12% greater uncertainty in the forecast results.






There is additional uncertainty in model results during the winter season, when ice forms in the Great Lakes.
Lake Ontario typically experiences some ice cover between December and April (Anderson et al., 2018),
which impacts lake processes, including water levels, circulation, and waves through limited air-water
momentum transfer (Anderson et al., 2018; Farhadzadeh and Gangai, 2017). While ice cover has been
simulated in Lake Ontario using other models (e.g., Oveisy et al., 2012), it is presently not available in
Delft3D-SWAN. Therefore, simulations of surface behaviour during the ice-covered months would have
limited accuracy in ice-covered areas. Future work could incorporate ice cover into the model or apply
dynamic masking of ice-covered surfaces using satellite data, to improve results during these months.

While this system requires low computational resources, making it possible to adapt it for other locations,
the applicability of the model is limited by the availability of online data for model forcing and validation.
In order account for seasonal changes in mean lake levels, near real-time measurements of water levels are
needed in the simulation to adjust the datum in post-processing. However, if no data were available the
simulation could include the wind-generated short-term fluctuations in surface levels and real-time
operations could continue. The workflow of the model is also limited by the availability of atmospheric
forcing data, with any interruptions of service in the HRDPS forecasts causing the hydrodynamic
simulations to fail for that run-cycle. Improvements in the system could account for this by providing a
secondary source of atmospheric forcing in that case. In future studies, we recommend applying this system
to a region in the coastal ocean, therefore requiring the development of real-time forecast inputs of open
boundary conditions.

**5 Conclusions**

A forecast model for wind-driven hydrodynamics was developed and applied to Lake Ontario using an
approach with relatively low computational demand. Wind-waves and water levels were simulated using a
dynamically coupled Delft3D-SWAN model driven by high resolution atmospheric forcing. Simulations
were able to forecast the wind-driven variability in the lake surface, with seasonal changes in the total water
levels accounted for by adjusting the datum for each forecast cycle based on observations of the mean water
level. The system provides rapid (~5 h runtime) predictions that are publicly available through the project
webpage, with the automated system forecasting a 48 h period every 6 h. The model has been running
continuously since April 2021, capturing a variety of storm events with storm surges up to 0.30 m and
significant wave heights over 4.00 m. Reliable prediction for wave conditions during winter months are



provided by the forecast model when no wave observations are available, however accuracy is limited
where ice is present as this process is not included in the modelling system.

Results show that the model is effective in simulating short term fluctuations in the water levels and wave
conditions during strong storm events, with relative errors between observed and forecasted storm surge
magnitudes and significant wave heights of less than 15%. Larger errors typically corresponded to locations
in the lake with larger ranges in observed water levels. For storm events, as the forecast lead time decreases
for progressing forecasts, the simulated results changed as a result of updates to the meteorological forcing.
No constant trends in forecast error due to decreasing forecast length were apparent, with forecast accuracy
increasing with shorter forecasts in some cases and staying constant at others, but overall results agreed
well with observed data for all forecasts leading up to an event, with RMSE for storm surge and waves
below 0.05 m and 0.30 m, respectively. The model compared well with other existing forecast models in
the Great Lakes (GLCFS), yielding comparable results for water level predictions during multiple storm
events. Due to the low computational requirements and pan-Canadian coverage from the High Resolution
Deterministic Prediction System forecasts, this model could be adapted to other Canadian lakes and coastal
seas with available bathymetry data for storm surge prediction and monitoring.

**6 Code and Data Availability Statement**

Real-time model results are available at https://coastlines.engineering.queensu.ca/lake-ontario/, and
archived on the server, to be made available by contacting the corresponding author. HRDPS input data
is available from the Meteorological Service of Canada Datamart and observed data is openly accessible
online, as cited in the text. The Python and MATALB scripts, and data used in this research is archived in
the Department of Civil Engineering at Queen's University and will be made available on
https://dataverse.scholarsportal.info/dataverse/queens upon manuscript acceptance. The open source
Delft3D software is available from Deltares (https://oss.deltares.nl/web/delft3d/ ).

**7 Author contributions**

The concept of the COASTLINES-LO workflow was designed by RM, LB, SS, and LS, and LS
implemented the idea. LS developed the performed the model simulations. All authors contributed to the



validation of the model and interpretation of the results. SL wrote the manuscript with contributions from LB, SS, and RM.

**8 Acknowledgments**

Funding for this research was provided by Natural Sciences and Engineering Research Council of Canada (NSERC) under the Discovery Grant program awarded to R.P. Mulligan (RGPIN/04043-2018), and a Queen's Dean's Research Fund award to L. Boegman, R.P. Mulligan and S. Shan.

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
