# Peer review of "Development and performance of a high-resolution surface wave and storm surge forecast model"

_Geoscientific Model Development, 2023_

## Author Response (AR1)

Geoscientific Model Development | Ms. Ref. No.: GMD-2023-151

**Development and performance of a high-resolution surface wave and storm surge forecast model (COASTLINES-LO): Application to a large lake**

Laura L. Swatridge, Ryan P. Mulligan, Leon Boegman, and Shiliang Shan

**Response to Reviewers Comments**

8 Feb, 2024

**Editor Comment:**

**I thank the referees for their comments. Based on these, I encourage Laura Swatridge and co-authors to respond to these comments and to prepare a revised manuscript for submission.**

**With good wishes,**

**Andy Wickert**

> **Response:** We thank the Editors and the anonymous reviewers for taking the time to read and provide comments and suggestions on how to improve the manuscript. All reviewer feedback has been addressed, by updating figures, clarifying details in the text, and improving the discussion of the results.
>
> This document provides a point-by-point description of modifications that were made to the manuscript based on the reviewer's feedback. These details are provided using indented blue text underneath each comment. Comments and responses from the first round of feedback are grouped together under the subheading 'Round 1 comments', then following comments and responses are under the 'Round 2 comments' heading.

**Round 1 Comments**

**Reviewer 1:**

> Response: As indicated by the Editor, the comments from Reviewer 1 were "unhelpful and self-serving". We therefore politely ignore these comments and respond in detail to the constructive comments from Reviewer 2 and Reviewer 3 below.

**Reviewer 2.**

Review of 'Development and performance of a high-resolution surface wave and storm surge forecast model (COASTLINES-LO): Application to a large lake' by L.L. Swatridge et al. submitted to the journal Geoscientific Model Development.

This is the coupled wave-current model's application to the Great Lakes, specifically Lake Ontario, and the parameters focusing on is the significant wave height and water level. The authors have done a lot of efforts in the numerical model development and the forecasting system, which is tested under both normal and storm conditions. The model performance has been validated by comparing with another popular forecasting system in the Great Lakes (i.e., Great Lakes Forecasting System, GLCFS) and NDBC (National Data Buoy Center) and ECCC (Environment and Climate Change Canada) observations. The authors conclude that their coupled 2D Deltft3D-SWAN model has a comparable ability with the GLCFS 3D FVCOM model, while it is more computationally efficient. Based on the reviewer judgement, the manuscript is interesting and nice, while it needs major revisions before being proceeding further. The following are my specific comments.

> Response: Thank you for the detailed review and suggestions on ways we can improve the manuscript. Your feedback is very much appreciated.

In 2 Method – 2.1. Modeling Approach. At Page 5 in Lines 135 – 138. 'The Delft3D simulations uses a curvilinear grid with a horizontal resolution gradually ranging from 250 – 450 m …… 350 – 600 m for the wave model'. Why do authors do not use the same grid for both the storm surge model and the wave model?

> Response: The wave model grid resolution was relaxed to reduce the computational requirements needed to run a simulation. When using the original grid (same as the circulation model), the computational time was increased to above 6 hours, as the number of computational cells is almost doubled in the higher resolution grid (333216 for circulation; 169400 for wave). To further justify this decision, the text has been updated as:
>
> Line 136: "The Delft3D simulation uses a curvilinear grid with a horizontal resolution gradually ranging from 250-450 m. The wave grid has a coarser resolution, ranging from 350-600 m, thus reducing the computational time required to complete a wave simulation while still achieving higher resolution in nearshore areas."

At Page 5 in Line 140, please add a space between '0.07' and 'm'.

> Response: Corrected

At Page 5 in Lines 144-146, 'Simulations use a time step of 120 s to satisfy ……'. Is the 120 s time step for the storm surge or wave model setting? What is the time step for another model?

Response: Text updated to add more details regarding time steps, as follows:

"Hydrodynamic simulations use a time step of 120 s to satisfy the Courant–Friedrichs–Lewy stability criterion, and the wave model uses a stationary computational approach"

At Page 6 in Fig. 1, please add the title of the colorbar, maybe 'Bathymetry (m)'?

Response: Figure updated with label on the legend.

At Page 6 in Lines 160-163: 'No lateral open boundary ……'. As far as the reviewer understand, the Niagara River is a river with larger river discharges. By not including it, the coastal circulation and wave dynamics maybe influenced. Could the authors show the influence or the difference by including the Niagara River (and the St. Lawrence River) for storm surge and wave simulations?

Response: While the Niagara and St. Lawrence rivers are the major inflows/outflows to the lake, we have concluded that including this influence in the model is not necessary, and outside of the scope of a real-time forecast model. Based on previous modelling studies in Lake Ontario (i.e. Prakash et al. 2007; McCombs et al. 2014), the influence of river flows only extends to approximately 10 km of the river inlet, thus for the large scale simulations in the current work, which focuses on lake-wide water levels and waves, this can be ignored, and is now justified in the text as:

Line 162: "No lateral boundary conditions are applied to account for the influence of the riverine flows (Niagara and St. Lawrence Rivers), as previous works have found the hydrodynamic influence of river flows is limited to within 10 km of the river inlet, and therefore have a negligible impact on large-scale circulation and water levels over event-based timescales (Prakash et al., 2007; McCombs et al. 2014a)."

The other major impact of the river flows is their influence on mean water levels in the lake. In the real-time system, this is included by updating water levels in the lake in the post processing stage based on observed data.

Line 190: "Seasonal changes in water levels due to inflows, outflows, and evaporation are not included, but are accounted for in post-processing.".

At Page 7 in Fig. 2, what do MDF and MDW stand for?

Response: Figure updated to say 'Model Definition Files' instead of MDF and MDW, to make workflow diagram more easily understood for readers

At Page 8 in Lines 201-203: 'Hourly surface waves and winds are measured in Lake Ontario at one US Natioal Data Buoy Center (NDBC) buoy and ECCC buoys ......'. Based on Table 1, it shows 3 NDBC + 1 ECCC buoys. Please double check and be consistent between the descriptions and table.

> Response: Thank you, table 1 was updated with the correct information, and checked to ensure this is consistent with the text.

At Page 9 in Table 1, why do the water levels stations add no information on the location depths?

> Response: Depth information for water level gauges is not available, likely as these stations are located around the perimeter of the lake in relatively shallow depths. We agree that this data gap in Table 1 is confusing, so to correct this, table 1 has been separated into two tables (one for wave buoys, one for water level gauges) and referenced in the text accordingly. Additional, the text describing the water level gauges has been updated:
>
> Line 203: Near real-time observations of water surface elevation ($\eta$) data are available at 9 water level gauges around the perimeter of Lake Ontario.

In the Method section, the reviewer considers that it is better adding the mathematical expressions for the statistics definition. For example, the (normalized) root mean square error, correlation coefficient, relative error etc.

> Response: Expressions for each error metric have been added in the text, and referenced accordingly

There are some mismatches between the texts and the Figure at Page 10. I suggest delete Fig. 3i in Line 228, change the Fig. 3c to Fig. 3e in Line 231. By doing so, the contents and the figure can be consistent with each other.

> Response: Thank you, text has been corrected as suggested.

At Page 10 in Line 229, please add 'the' between 'overpredict' and 'maximum'.

> Response: Corrected

At Page 11 in Lines 245-246, based on the Fig. 1, I think station 'East Lake Ontario' in the east of the lake not northeast.

> Response: Yes, that's true. The text has been updated as:
> "Stations in the eastern end of the lake (Prince Edward Point, East Lake Ontario)…"

At Page 11 in Lines 245-247, 'Stations in the northeast region of the lake ...... generally experienced the largest waves, due to the prominent northeasterly direction of storms over the lake resulting in a larger fetch at these locations.' Could you show me the wind map and time series?

Response: Reference have been added to support the statement describing the dominant wind patterns over Lake Ontario (ie. Lacke et al. 2007; McCombs et al. 2014a). In addition, the reviewer can refer to Figure S2 in the supplementary material for an example of a wind field/time series validation of wind speeds over Lake Ontario for the first selected storm event.

Lacke, M. C., Knox, J. A., Frye, J. D., Stewart, A. E., Durkee, J. D., Fuhrmann, C. M., & Dillingham, S. M. (2007). A climatology of cold-season non convective wind events in the Great Lakes region. Journal of Climate, 20(24), 6012-6022. https://doi.org/10.1175/2007JCLI1750.1

At Page 12 in Fig. 4, why the simulations have a data gap in about Feb. 2022?

Response: Thank you, good catch. The model was offline for a period between February 9 – 27, 2022 as a result of a service change in the HRDPS meteorological system. The modelling system had to be updated to account for the new delivery format for the atmospheric inputs. This explanation has been added into the Figure 4 caption as follows:

"Note that the model was offline and are unavailable between February 9 – 27, 2022 due to a change of service for the meteorological inputs."

At Page 13 in Section 3.2. Storm event forecasts. The authors select the November 11, 2021, storm event to check the model performance. Why the authors choose this event to study? In addition, it would be better to choose more storm events to examine the model performance under storm conditions, e.g., more than 2.

Response: This event was selected as this is the largest event with available observed data at all wave buoys, thus allowing for the most complete validation possible. The second event selected had limited available wave data but was the strongest event over the operational period. Both events had distinct wind fields, thus representing model results over a wide range of conditions. The text has been updated as:

Line 280: "The performance of the model was evaluated over an event on November 11, 2021, which generated the largest waves and storm surge over the 20-month operational period with available observed water level and wave data."

In the scope of this paper, the authors feel analysis of these events is sufficient to validate model performance. However it is also noted that in the long time series comparison of results many storm events are included, and captured with good agreement.

We do agree that further investigation into storm events would be valuable and suggest this as a further recommendation.

Line 469: "Additional investigation of real-time model performance during storm events when the lake is stratified is recommended for further model validation."

At Page 13 in Lines 271-272, 'A setdown of about 0.10 m was recorded at the Burlington station, which was underpredicted by the model by up to 0.05 m'. The under-prediction for this station is large (e.g., 50%), could they explain the reason for this bias and can it be improved? The reviewer is not sure that why the 50% error here, but it is 0-20% error in Figure 6c for the same station and event?

Response: Yes, the relative error at this station that was computed remains below 20% over the storm duration, despite a maximum observed and modelled set down of 10 and 5 cm. . This is because the calculation of relative error in Figure 6 is in reference to the mean water level in the lake at the beginning of the event, not to the zero datum. We agree this is unclear and creates misleading error statistics in Figure 6, but this decision was made to allow for consistency in storm surge values for consecutive forecasts. To clarify this, the relative error mathematical expression has been added, as referenced above to show how this is calculated (Eq. 4). In addition, the text has been updated:

Line 226: "For each forecast, the relative error (RE; eq. 4), between observed and simulated maximum storm surge relative to the mean water level at each station, or wave heights, was computed…"

Figures 5, 7, 9, 10 needs improvements since there are so many solid lines in one figure, which make the reviewer hard to identify it.

Response: We agree that these figures show too many lines, making them difficult to understand. To improve the plots, the number of overlapping forecasts in Figures 5 and 7 has been decreased, from 16 forecast to 9, and the x-limits of the figure have been reduced so the forecasts of the storm event are larger and easier to see.

Figure 9 has been updated in a similar way, now only showing 8 forecasts over a shorter time period.

In Figure 10, all forecast lines were kept, as each one corresponds to a point on the scatter plots on the right, and therefore we feel it is important to show all the information. To improve the clarity of this plot, the colormap has been updated to improve the contrast between the different forecasts, and the limits of the plot have been shortened.

At Page 12 in Figure 4, the authors show the comparison of the Hs between the simulation and observations. How about the peak wave period (Tp)?

Response: Plots showing period results are now included in Figure S2 in the supplementary material, and referenced in the text as:

Line 258: "Results showing forecasted wave period compared to observations are shown in Fig S2 in the supplementary material."

Figures 6, 8, 10: Usually, the RE and RMSE could be improved as the length time for prediction decreases, why not all points follow this trend? For example, could the authors explain Page 19 Lines 360-361 'However, after the 18 h forecast there was a slight increase in RE from less than 1 % to about 5 % (Fig. 10b)'?

Response: We agree that that would be the expected trend, based on how atmospheric predictions tend to increase in accuracy with reduced lead time. However, there are other factors contributing to uncertainties in the results, such as model resolution, initial conditions, and background hydrodynamic processes that are not included/resolved in the model.

We note that the change from 1% error to 5% error observed in Figure 10b only corresponds to about 5 cm difference between forecasted water levels. The increased RE may be due to error in the magnitude and direction of the wind fields. This could also be an issue with the model setup, as calibrated parameters (i.e. friction, viscosity) can influence results. These were tuned to try to achieve optimal performance for a range of conditions, however it is not feasible to be able to account for all possible storm conditions. As this is a real-time model, additional calibration/adjustments could not be made to improve results for specific events, and the response of the lake to the unique conditions for each storm event is different.

Some discussion on this was added: Line 402: "Cases where error increases (i.e.. Fig 10b) or remains constant (i.e. Fig. 8), can be explained due to sources of uncertainty in the model calibration and neglecting additional hydrodynamic processes in the model setup (i.e. 3-dimensional circulation).

At Page 15 in Lines 300-302, 'Measured waves during …… due to the shift in wind direction during the storm'. Could the authors more specifically point out the wind direction shift from which to which?

Response: The text has been updated to give a more detailed description of wind direction over the event:

Line 321: "Measured waves during this event reached up to 2.10 m, with the buoys in the western region of the lake (Fig. 7c, d) experiencing peak wave heights about 12 h earlier than the buoys in the eastern region of the lake (Fig. 7a, b). This is explained by the shift in wind direction over the storm duration, with winds originally from the southeast, rotating clockwise, then blowing dominantly from the west along the axis of the lake (Fig. S2 in the supplementary material) ."

At Page 17 in Figure 8, what do these arrows stand for in panels a and b?

Response: Figure caption updated to add additional details explaining the plot, as follows:

"Contour plots showing maps of modelled waves with vectors indicating wave direction at the peak of the storm event from two forecasts, starting a) November 11, 00:00 UTC and b) November 12, 00:00 UTC with observed data plotted at the observation locations in black circles. Note that every 10th vector is plotted for clarity.|"

At Page 19 in Line 347, 'select stations' maybe changed to 'selected stations'?

Response: Corrected

At Page 23 in Line 455, pleas add 'to' after 'In order'.

Response: Corrected

The authors emphasis that the Delft3D-SWAN (COASTLINES-LO) is highly computational efficient and can be easily applied to other lake systems. The reviewer would suggest they add a Table to compare the computational information between their system and GLCFS (e.g., computational nodes, elements, time step, total time, computational cores, parameter information etc.)

Response: A table summarizing the key differences between the modelling systems has been added to the supplementary material, and references in the text as follow:

Line 411: " Differences between predictions from these models can be explained according to the setup of each system, including different hydrodynamic models, grid resolutions, and atmospheric forcing inputs, which are summarized in table S2 in the supplementary material."

**Reviewer 3:**

I am happy to provide feedback on this manuscript. The manuscript presents a script written in Python/Matlab that performs pre-processing, running, and post-processing of a depth-averaged Delft3D+SWAN model to forecast water levels and waves in Lake Ontario for 48 hours. The manuscript is well-written and easy to understand. This study has the potential to make a valuable contribution to water management in Lake Ontario. However, the scientific/operational contributions of the proposed modeling framework and better discussion would benefit from improvement. Therefore, I recommend a "moderate revision" of this manuscript before it is published.

Please find below some specific comments:

The term "automated prediction" is used ambiguously in the abstract and several parts of the manuscript. It is recommended to provide a detailed explanation or use a different term altogether.

Response: The term "automated" in the abstract has been removed, and rephrased as "A real-time forecast model of surface hydrodynamics in Lake Ontario (Coastlines-LO) was developed to automatically predict storm surge and surface waves…"

We feel the use of the phrase throughout the manuscript accurately describes how the prediction system continuously runs without the need for any human inputs. Section 2.2 has been updated to make it more clear that all workflows are automated:

Line 173: "For pre-processing, initiation of the modelling system is scheduled to occur when a new HRDPS forecast becomes available"

The reason for the low computational demand of the proposed modeling framework is due to the lower spatial resolution used in the Delft3D/SWAN grid, as well as the depth average configuration which turns the 3D model into a 2D model. Additionally, the GLCFS is currently operational on NOAA's computational system, which means that its computational cost is affordable. Therefore, it is important to clearly state the operational and scientific contributions of the proposed modeling framework.

Response: Yes, to run this system on the local computers, the resolution was limited and certain processes had to be neglected. Despite this, results compare well with the operational system developed by NOAA. We have highlighted the novelty of this finding in the discussion and conclusion sections.

Please provide the source for the measurements and observations mentioned in Figure 1.

Response: A reference to table 1 has been added in the caption for Figure 1, to provide information about the observation points. The bathymetry source has also been added in the caption, and we refer the reviewer to Line 142 for a detailed description of the bathymetric dataset.

Could you confirm if the Python/Matlab scripts are currently being used for operations? Also, are these scripts available to the public?

Response: Yes, the model is currently operational, and the results are updated in real time on the project webpage: (https://coastlines.engineering.queensu.ca/lake-ontario/).

All model scripts, and input files, as well as results referenced in the manuscript are archived on Zenoda and made available for viewing through the link: (https://doi.org/10.5281/zenodo.10407863, Swatridge, 2023), as described in section 6. Code and Data Availability Statement.

It is suggested to also include metrics such as RMSE and RE (%) in Figure 11, and to add a table to clearly indicate the differences between the proposed model and GLCFS.

Response: A table summarizing the key differences between the modelling systems has been added to the supplementary material, (Table S2) and references in the text.

Summary error statistics from the comparison of the two models has been added in table S3 in the supplementary material and is referred to in section 4.2.

It is recommended to include the different viewpoints and angles in the proposed modeling framework.

Response: Through the development of this modelling system, a balance between computational efficiency and accuracy had to be achieved to allow the model to run in the required timeframed. We recommend in future work to expand the analysis to include an investigations the effects of including ice in the model , or different wind field inputs, and applying the modelling system to other large water bodies with open boundaries where connected to the ocean.

**Reviewer 2:** https://doi.org/10.5194/gmd-2023-151-RC14

**RC 14:** Thanks for your point-by-point reply to my comments. In general, I would first suggest authors uploading a new version of the manuscript and its with track of change, so that I could check what changes they have made during the first round of the review. This manuscript needs at least substantially major revisions before being reviewed again. The following is my concern to their reply:

> Thank you for the thoughtful feedback. The updated manuscript will now be uploaded for viewing now that the initial open discussion has ended. Hopefully the revised manuscript will help addressing the reviewer's concerns.
>
> In addition to the initial responses to your comments, the manuscript has been updated according to your additional feedback, as outlined below.

1. It can be added a Table to explain it more clearly. What is the original grid number and calculation time for the circulation model and wave model, and these information after the wave model is relaxed? Why not relaxing both models? For the modeling paper, I consider it is necessary to show us in a more detailed way of the model grid, e.g., showing a map.

   **Response:** The information has been compiled in a table, and inserted into the supplementary material, as Table S4.

   Table S2: Approximate runtimes for model resolution configurations using a 16-cores of a XEON 2.50 GHz processor workstation with 64 GB RAM, to simulate 48 hrs of model time. Note that the simulation run times are approximation, as this value changes depending on the conditions being simulated, which impacts how long it takes for the wave model to converge on a solution to the required confidence criteria.

   | Configuration | Flow Resolution | Flow grid cells | Wave Resolution | Wave grid cells | Runtime |
   |---|---|---|---|---|---|
   | 1 | 250m – 450m | 333216 | 350m - 600m | 169400 | ~4 hrs |
   | 2 | 250m – 450m | 333216 | 250m – 450m | 333216 | ~7 hrs |
   | 3 | 250m-450m | 333216 | No Waves | N/A | ~0.5 hrs |

   From this table, should be noted that the computational requirements to run a flow simulation are much smaller than a wave simulation, as a stand alone flow run only takes about 30 minutes for a 2 day forecast. Therefore, the decision to reduce the resolution of the wave model but not the circulation model allows for the resolution to be preserved in the water level simulation without impacting the runtime in a significant way. This is important for achieving accurate results, especially in regions like the Kingston basin where more complex coastal features are present, and the higher resolution helps to resolve this.

A detailed description of the model setup, including visual description of the grid can be found in a related study (Swatridge et al. 2022). This is referenced in the text at Line 134.

3. The authors still do not show us the time step for the wave model. Even for the stationary mode, it still has the time step to calculate. While is it fine to use the stationary mode for the realistic forecast with a lot of variables changing (e.g., wind field)?

**Response**: The stationary wave model has a time step of 60 minutes, the same as the coupling interval. Each stationary simulation assumes waves reach fully developed conditions within the defined time frame. Therefore, over that one hour period, the wave model doesn't move forward in time (however the flow model computations continue with a smaller time step), but the equations are iterated until the solution reaches the convergence criteria defined by the user. In the current work, the accuracy criteria is set at the default values of a relative change of 0.02 m in significant wave height between iterations for 98% of wet grid cells.

This has been rephrased as: Line 146 "Hydrodynamic simulations use a time step of 120 s to satisfy the Courant–Friedrichs–Lewy stability criterion, and coupling with the stationary wave model occurs every 60 minutes."

The use of a stationary models gives the advantage of improving model stability (Rey et al. 2020), which is the main benefit for use in a real-time forecasting system. However, we also acknowledge that the assumption of instantaneous wave propagation across a domain, leading to fully developed conditions may be a source of error (Camerena and Gutier, 2013). Therefore, stationary runs are typically recommended when the waves have a relatively short residence time in the computational area compared to the time scale of changes in wave boundary and forcing conditions. However, Sheng et al. (2010) used a stationary SWAN model to simulate hurricane wave fields and found comparable results between the two computational modes.

References:

Camarena, A., & Gautier, C. (2013). Comparison stationary vs non-stationary SWAN runs, Wadden Sea hindcast 5/6 Dec 2013. Deltares.

Rey, A. J. M., Corbett, D. R., & Mulligan, R. P. (2020). Impacts of hurricane winds and precipitation on hydrodynamics in a back barrier estuary. Journal of Geophysical Research: Oceans. 125(12).

Sheng, Y. P., Alymov, V., & Paramygin, V. A. (2010). Simulation of storm surge, wave, currents, and inundation in the Outer Banks and Chesapeake Bay during Hurricane Isabel in 2003: The importance of waves. Journal of Geophysical Research: Oceans, 115(C4).

5. Not satisfied. The reviewer asks for authors showing the hydrodynamic results with and without two major river flows into Lake Ontario, while they avoid doing so. And indicate that it is not important for their forecasting system. Coastal circulation is one important part for the lake circulation, and the reviewer holds the opinion that major river flows better being considered.

**Response:** Daily flows are available online in near real-time, river flow forecasts are not, which complicates using inflows and outflows in a forecast model. Therefore, commonly they are neglected in forecasting models (e.g., in Lake Erie forecasting, Lin et al 2022, GMD). We agree that river discharges impact the lake local circulation, as the inflow and outflows can create cyclonic eddy patterns and stronger flows near the inlets/outlets (Hui et al. 2021). We, therefore, expect

errors in simulated currents near the Niagara River inlet and the St. Lawrence River outlet, because we neglect these flows. For detailed studies of lake current patterns, this consideration would be essential. However, given that we are already using a depth-averaged circulation model, as the goal of the current work is to develop a system for storm surge and wave prediction, we feel that trying to accurately resolve nearshore currents is beyond the scope of the present work. Rather, we need a simple model that can run quickly and efficiently, with a priority on simulating water levels, as opposed to currents, and so it was decided to neglect river boundary flows.

Closed basin modelling approaches have been implemented in Lake Ontario (McCombs et al. 2014), Lake Winnipeg (Chittibabu and Rao, 2012), and Lake Michigan (Mao and Xia, 2017), which were able to achieve accurate water level results. In addition, the influence of including and not including river boundaries was investigated in a numerical modelling study in Lake Ontario by Prakash et al. (2007), who determined the area of influence of the Niagara River extends to about 10 km from the mouth, and determined for lake wide circulation patterns, the open and closed system performed similarly.

Therefore, we feel the decision to treat Lake Ontario as a closed system is justified through the referenced previous works and acknowledge that this simplification does generate some uncertainties for results in the river regions. This is acknowledged in the text as:

Line 167: "The closed based approach leads to uncertainties in the simulated results in the river region, however the impacts on the lake-wide hydraulics is expected to be minimal."

References:

Chittibabu, P., & Rao, Y. (2012). Numerical Simulation of storm surges in Lake Winnipeg. Natural Hazards, 60, 181-197. https://doi.org/10.1007/s11069-011-0002-7.

Hui, Y., Farnham, D. J., Atkinson, J. F., Zhu, Z., and Fend, Y. (2021). Circulation in Lake Ontario: Numerical and Physical Model Analysis. *J. Hydraul. Eng.* 147(8).

13. Not satisfied. First, this expression is ambiguous of the time period for the storm event, it is needed to mention during which time period for the largest winds and waves. Moreover, by referring to the right spatial maps of winds in Figure S2, neither of them shows northeasterly winds as stated by the texts.

    **Response**: The statement describing the dominant wind patterns over Lake Ontario doesn't refer to a specific event or time period, instead is a broad characterization of the wind environment over the Great Lakes. This is backed up by past works, which have found non-convective or extra-tropical systems that originate in the southern and central Rockies, Canada, or the gulf tend to move northeast over the region, with the strongest winds towards the north or east directions (FEMA, 2007; Lacke et al. 2007). Storms with this characteristic wind patterns have been studied in other modelling works in Lake Ontario, finding that this direction results in the largest impacts due to the orientation of the lake. (Swatridge et al. 2022; McCombs et al. 2014).

    The text has been updated to clarify that this statement is not referring to a specific event:

Line 261: "Stations in the eastern end of the lake (Prince Edward Point, East Lake Ontario) are expected to experience the largest waves due to the experienced the largest waves, due to the prominent north-easterly direction of storms over the lake, which results in winds blowing along the long-axis of the lake creating a resulting in a larger fetch at these locations (Lacke et al. 2007; McCombs et al. 2014a)."

We agree that the wind fields for storm 1 do not clearly show this pattern, instead rotating clockwise over the lake during the events. At times t2 and t3 in Figure S3, the winds shift direction from North to East, giving an example of how this wind direction results in longer fetch, which is described in the text in detail in section 3.2.

In addition, the wind fields for event 2 have been added in the supplementary material (Figure S4), demonstrating these wind patterns. This figure is referenced in the text:

Line 369: "Water level forecasts during a storm event on December 8, 2021, were examined in relation to forecast lead time. During this event, 21 m s$^{-1}$ winds (Figure S4 in the supplementary material)"

14. Please be careful when revise the manuscript, "...... due to a change of is delivery format ......", it is obviously wrong in grammar. Please revise it.

    **Response**: This has been updated to: "a change in its delivery format'.

15. Based on Figure 4, it seems the largest wave event is around 10-Dec-2021, not November 11, 2021 as described by the authors. Please double check it. Again, why not choosing more than 2 events for this study?

    **Response:** We agree the initial phrasing of this was unclear. Yes, December 10$^{th}$ had the largest waves in the validation period, however no observational data is available at that time, as buoys are removed from the lake in the winter. The November 11$^{th}$ event is the largest storm where all buoys where in the lake, so there is available measured data for comparison. This has been reworded in the text to more clearly explain the decision:

    Line 282: "This event was selected due to the large storm generated ($\eta$ = 0.17 m), and it resulted in the largest waves over the 20 month operational period in which available observed water level and wave measurements are available from all buoy locations for comparison."

    As to the decision to include only 2 events for analysis, the authors believe that since Figure 3 and 4 display long term results of water levels and waves, compared to observations, these provide a long term and 'big picture' overview of how the model performs under varying conditions (light wind conditions, varying wind directions, and multiple storm events are captured). The general analysis of this data proves model ability to achieve good results over a wide range of wind scenarios.

    To be concise in the manuscript, the decision was made to focus in detail on a select 2 storm events, chosen due to the extreme conditions for each storm. This allows for a detailed analysis of all aspects of the model performance during these events in the text.

17. It is still unclear to me. And I think 50% error is too large and the model needs to be improved for storm surge prediction skill.

When considering the maximum absolute error at this location, it is only 5 cm. This value is compared to performance metrics for other existing forecast models and falls within acceptable model performance metrics. For example, in the performance report for NOAA's Great Lakes Coastal Forecasting System, acceptable magnitude of error for water level extreme events is set at +/- 15 cm (Kelley et al. 2018). Therefore, the 5 cm difference is acceptable by these standards.

We agree that 50% error is extreme, however as noted in the initial comment, the 5 cm difference in the first forecast (-5 cm) and observed peak water levels (-10 cm), that is referred to, is displayed as a relative error of ~20%, not 50%. This is because the observed data vertical datum is converted to match the results using the average water level at each station, which is dynamically changing. This results in the mean water level, at each station, not starting at the 0 shown in the plots, but instead above or below this point. Note that at the Burlington station in Figure 5h, the initial water level for observations and modelled results is around 7 cm. This same datum is used in all calculations for this event at each station, to ensure consistency in the results.

19. I cannot find the added peak wave period comparison.

The revised supplementary material has now been uploaded. Please see Figure S2.

20. I am still very confused with the explanation. The west winds are after November 12 (e.g., 11/12 16:00), why they are used to explain the results before November 12 (e.g., 11/11)?

Response: Thank you for pointing this out, the text has now been updated to convey the analysis that was performed during this event in a clearer way. The storm involved sustained winds over a 24-hr period, in which the direction rotated from towards the west, then north, then dissipating as the winds blew towards the east. I believe much of the confusion here is due to ambiguity in the language relating to if the winds are 'coming from' or 'going towards' the west. To resolve this communication issue, the following update was made:

Section 3.2. Line 280: "…, consisting of wind speeds that approached 15 m s$^{-1}$, with the direction rotating clockwise from blowing towards the northeast to the winds dominantly blowing towards the east over a 24 h period."

Potential uncertainty in the interpretation of these results may also be due to the information shown in Figure 8. This is comparing the lake at the same time (November 12, 2021), with forecast results that started on a) November 11, 00:00 UTC, and b) November 12 00:00 UTC, those showing results for a 32-hr forecast, and a 8 hour forecast. To clarify this comparison, Figure 8's caption has been updated:

"Figure 8: Contour maps of modelled waves with vectors indicating wave direction at a select time during the storm event from two forecasts, with an a) 32 hr lead time starting November 11, 00:00 UTC and b) 8 hr lead time starting November 12, 00:00 UTC…

A similar change was made for the caption to Figure 6.

I go to the forecast website (https://coastlines.engineering.queensu.ca/lake-ontario/), while it only shows the station location and information, I could not find the realistic and forecast simulations of winds, waves and water levels on this website.

> Thanks for checking it out! We have checked the modelling system, and it appears to still be actively updating online as of the revision submission date (April 5, 2024).
>
> We note that some plots don't load instantly and take a few seconds to appear. We also suggest navigating through different tabs in the dop down menus. Here is a screenshot from today, after clicking on the 'Forecast Results" tab.

[Figure]

**RC8:** Thank you for your effort in revising the manuscript. Overall, the authors have responded well to most of my comments.

However, I am still not entirely convinced about the novelty of the proposed workflow. For instance, it is unclear why running the model on a desktop PC is essential for operational applications, and there is no

evidence that this workflow can be easily adapted for different lakes. Therefore, I kindly request you reconsider this reply and modify the manuscript accordingly.

> **Response:** Thank you for the response, we have made an effort to more clearly convey the novelty and advantages of this modelling system and approach. Indeed, our group has already applied the same workflow for Lake Erie, coastal North Carolina, and the Bay of Fundy. We cite much of this work, providing evidence that the workflow is easily adaptable. The need for running in a desktop PC, is that not all users have access to high performance multi socket multi core servers to run CFD code. Therefore, to truly be flexible and adaptable, the workflow must also be computationally efficient.
>
> The development of operational forecast system that can balance computational demand and accuracy is outlined as a key direction for future work in the coastal research community by Elko et al. (2019). We feel that the use of local computing resources is a key indicator of model efficiency. This motivation has been added into the text at:
>
> Line 66: "This need to effectively balance efficiency and accuracy in real-time models is an active research area (Elko et al., 2019)."
>
> We feel that the open source and low computational demand of the system is one of the key features that provides novelty to this work. The results aim to advance the ability to apply numerical models in real-time to other coastal regions by demonstrating that a simple approach can achieve comparable results with established operational models. The use of local computing resources acts as a good indicator of model efficiency when comparing to other operational systems.
>
> In Section 4.2, the importance of this comparison has been updated:
>
> Line 447: "This demonstrates that a relatively simple modelling system can be applied to coastal environments to achieve accurate and efficient hydrodynamic predictions. The open-source and flexible wrapper code could therefore be theoretically adapted to include different hydrodynamic models and investigate different field sites as previous works have successfully applied similar approaches for forecast modelling (e.g., Lin et al., 2022; Rey and Mulligan 2021)."
>
> These finding can be used in the development of future and existing forecasting systems, aiming to develop more efficient or flexible modelling systems, which is discussed in section 4.3.
>
> Line 482: "While this system requires low computational resources, making it flexible for adaption to other coastal regions, it's capability for forecasting in additional locations is an area that requires future investigation."
>
> Elko, N., Dietrich, C., Cialone, M.A., Stockdon, H., Bilksie, M. W., Boyd, B., Charbonneau, B., Cox., D., Desback, K., Elgar, S., Lewis, A., Limber, P., Long, J., Massey, C., Mayo, T., McIntosh, K., Nadal-Caraballo, N.C., Raubenheimer, B., Tomiczek, T., Wargula, A. E. (2019). Advancing the understanding of storm processes and impacts. Shore & Beach, 87(1).

---

## Author Response (AR2)

Geoscientific Model Development | Ms. Ref. No.: GMD-2023-151

**Title: Development and performance of a high-resolution surface wave and storm surge forecast model: Application to a large lake**

Author(s): Laura L. Swatridge et al.
MS type: Model experiment description paper
Iteration: Major revision

**Response to Reviewers Comments**

**Public justification (visible to the public if the article is accepted and published)**:
Dear Ms. Swatridge & co-authors;

I have now received two referee reports. The first indicates a desire to publish your manuscript essentially as-is. The second suggests rejection of your manuscript. This latter referee notes several points which, at least to me, look more straightforward to address than would be suggested by the "reject" suggestion indicated here.

I would ask your team to address Referee #2's eight points by responding to them and updating your manuscript and/or supplementary products, as appropriate, accordingly. I believe that you can improve the quality and impact of the manuscript through this process. I look forward to seeing both these responses and the updates that you make to the text, figures, and additional information.

Best wishes,
Andy Wickert

> **Response:** We thank the editor and reviewers for their time in continuing to review this manuscript, and are optimistic that through addressing the comments, the manuscript has greatly improved in quality and clarity throughout this process. Please find below a response to all comment from the referee reports. These details are provided using indented blue text underneath each comment.

**Report 1:**

> **Response:** No comments to address. We thank the reviewer for carefully reviewing the manuscript.

**Report 2:**

Suggestions for revision or reasons for rejection (visible to the public if the article is accepted and published) Reviewing of the round 2 version of the manuscript 'Development and performance of a high-resolution surface wave and storm surge forecast model: Application to a large lake' by L.L. Swatridge et al. (2024).

Thanks for the authors of their responses to my comments. However, the revisions and comments are not substantial to improvement of the manuscript. My suggestion is Reject.

The following are specific comments.

1. For a research paper, I could not clearly find the research gap from previous studies and the novelty of this work. For example, what the new knowledge could we gain from this study? This should not just a case study and model application analysis.

   **Response:** This is not a research paper, it was submitted as a model experiment description paper. We believe the findings of this manuscript fit within this scope, and have demonstrated that the application of Delft3d-SWAN as a real-time forecast model in the Great Lakes is novel.

2. I was thinking the modeling results based on different model grids while the authors have not given. For example, how the grid changes of wave model influence the simulation results of water level, storm surge, and wave statistics?

   **Response:** we agree that model resolution is a fundamental aspect of model accuracy, as values for storm surge and wave statistics will change as more details of the coast and small scale hydrodynamic features are resolved, as has been shown in previous works (eg. McCombs et al. 2014; Bastidas et al. 2015).

   The model resolution is comparable to other studies that have achieved accurate results in the great lakes in hindcast studies (Mao and Xia, 2017; Lin et al. 2022). Therefore, the use of this grid in real-time capacity is justified. In addition, the results are compared to the GLCFS model, which has a higher resolution in coastal areas (up to 30 m), and it is shown that the results are comparable (Section 4.2).

   Bastidas, L. A., Knighton, J., and Kline, S. W. (2016). Parameter sensitivity and uncertainty analysis for a storm surge and wave model. *Natural Hazards Earth System Science.* 16. 2195-2210.

   Lin, S., Boegman, L., Shan, S., and Mulligan, R.P.: An automatic lake-model application using near real-time data forcing: Development of an operational forecast workflow (COASTLINES) for Lake Erie, Geosci. Model Dev, 15(3), 1331-1353, https://doi.org/10.5194/gmd-15-1331-2022, 2022.

   Mao, M., and Xia, M.: Dynamics of wave-current-surge interactions in Lake Michigan: A model comparison, Ocean Modelling, 110, 1-20, https://doi.org/10.1016/j.ocemod.2016.12.007, 2007.

   McCombs, M.P., Mulligan, R.P., Boegman, L., and Rao, Y.R.: Modelling surface waves and wind-driven circulation in eastern Lake Ontario during winter storms, J. Great Lakes Res., 40(3), 130-142, https://doi.org/10.1016/j.jglr.2014.02.009, 2014a.

3. Still, I would require the authors show the modeling results by including the river discharges, since they could be important to the lake hydrodynamics in a relatively shallow lake.

**Response**: We have already responded to this.  It is not feasible to include inflows and maintain a water balance over long term simulations. This will not impact storm surge and surface waves, which we discussed in our last response. It will impact currents near the Niagara river inflow, but we do not seek to model/forecast currents.   This lake is ~200 m deep, it is not shallow as the reviewer claims.  This is beyond the scope of the present work.

4. For the storm event selection, it is ambiguous to me what are the differences between 'largest event' and 'strongest event'?

   **Response:** The definitions of what is considered the 'Largest events' are explained in the text:

   Line 282: This event was selected due to the large storm surge generated ($\eta = 0.17$ m), and it resulted in the largest significant wave height that occurred over the 20 month operational period with wave measurements available at all buoy locations for comparison.

   Line 349: For further investigation into model performance during storm events, wave forecasts during the event that resulted in the largest observed wave heights (December 1, 2022, Fig. 3c) were examined

5. 50% relative error for the storm surge prediction is not satisfactory, it needs to be improved.

   **Response**: As this is a real-time operation model, results cannot be calibrated for every possible event, therefore greater uncertainty is expected in these results compared to hindcast models.

   As we have noted, this error (shown in Figure 5h) corresponds to an absolute error of 5 cm, which falls within acceptable performance ranges developed by NOAA (+/- 15 cm), as discussed in the previous response (Kelley et al. 2018). The is also compared to acceptable performance criteria for hydrodynamic models developed by Williams and Esteves (2017), which defines a maximum error in peak water level of 15 cm as acceptable.

   We also note that as the forecast lead time decreases the error improves, with a minimum error between observation and modelled results at this location of 2 cm. This is updated in the text as:

   Line 293: "A set down of about 0.10 m was recorded at the Burlington station, which was underpredicted by the model by up to 0.05 m for the initial forecast results, and improves as the forecast lead decreased to a prediction of 0.08 m."

   Williams, J. J., and Esteves, L. S. Guidance on setup, calibration, and validation of hydrodynamic, wave, and sediment models for shelf seas and estuaries. *Advances in Civil Engineering.* 2017(1).

6. Figures 5,7,9,10 have so many solid lines, which is difficult to identify. The quality of this figure is not acceptable.

   **Response:** This is a personal opinion.  The other reviewer and the authors find this figure acceptable. We argue that the multiple overlapping lines are an important demonstration of evolving model forecast.

7. While my comments suggest authors to compare their forecast systems with GLFS, they select to ignore my comments.

      **Response:** See section 4.2 in the discussion where the model is compared to the Great Lakes Coastal Forecasting System (GLCFS)

8. Why the forecasting system (https://coastlines.engineering.queensu.ca/elementor-2122/) of wind speed is March 2024 not June 2024?

      **Response:** This is a bug in a website and has nothing to do with the submitted MS.